# Practical Contextual Bandits with Feedback Graphs

**Mengxiao Zhang** *
University of Southern California
mengxiao.zhang@usc.edu

**Yuheng Zhang** *
University of Illinois Urbana-Champaign
yuhengz2@illinois.edu

**Olga Vrousgou**
Microsoft Research
Olga.Vrousgou@microsoft.com

**Haipeng Luo**
University of Southern California
haipengl@usc.edu

**Paul Mineiro**
Microsoft Research
pmineiro@microsoft.com

## Abstract

While contextual bandit has a mature theory, effectively leveraging different feedback patterns to enhance the pace of learning remains unclear. Bandits with feedback graphs, which interpolates between the full information and bandit regimes, provides a promising framework to mitigate the statistical complexity of learning. In this paper, we propose and analyze an approach to contextual bandits with feedback graphs based upon reduction to regression. The resulting algorithms are computationally practical and achieve established minimax rates, thereby reducing the statistical complexity in real-world applications.

## 1 Introduction

This paper is primarily concerned with increasing the pace of learning for contextual bandits [Auer et al., 2002, Langford and Zhang, 2007]. While contextual bandits have enjoyed broad applicability [Bouneffouf et al., 2020], the statistical complexity of learning with bandit feedback imposes a data lower bound for application scenarios [Agarwal et al., 2012]. This has inspired various mitigation strategies, including exploiting function class structure for improved experimental design [Zhu and Mineiro, 2022], and composing with memory for learning with fewer samples [Rucker et al., 2022]. In this paper we exploit alternative graph feedback patterns to accelerate learning: intuitively, there is no need to explore a potentially suboptimal action if a presumed better action, when exploited, yields the necessary information.

The framework of bandits with feedback graphs is mature and provides a solid theoretical foundation for incorporating additional feedback into an exploration strategy [Mannor and Shamir, 2011, Alon et al., 2015, 2017]. Succinctly, in this framework, the observation of the learner is decided by a directed feedback graph $G$: when an action is played, the learner observes the loss of every action to which the chosen action is connected. When the graph only contains self-loops, this problem reduces to the classic bandit case. For non-contextual bandits with feedback graphs, [Alon et al., 2015] provides a full characterization on the minimax regret bound with respect to different graph theoretic quantities associated with $G$ according to the type of the feedback graph.

However, contextual bandits with feedback graphs have received less attention [Singh et al., 2020, Wang et al., 2021]. Specifically, there is no prior work offering a solution for general feedback graphs

---

*Equal contribution.

37th Conference on Neural Information Processing Systems (NeurIPS 2023).

and function classes. In this work, we take an important step in this direction by adopting recently developed minimax algorithm design principles in contextual bandits, which leverage realizability and reduction to regression to construct practical algorithms with strong statistical guarantees [Foster et al., 2018, Foster and Rakhlin, 2020, Foster et al., 2020, Foster and Krishnamurthy, 2021, Foster et al., 2021, Zhu and Mineiro, 2022]. Using this strategy, we construct a practical algorithm for contextual bandits with feedback graphs that achieves the optimal regret bound. Moreover, although our primary concern is accelerating learning when the available feedback is more informative than bandit feedback, our techniques also succeed when the available feedback is less informative than bandit feedback, e.g., in spam filtering where some actions generate no feedback. More specifically, our contributions are as follows.

**Contributions.** In this paper, we extend the minimax framework proposed in [Foster et al., 2021] to contextual bandits with general feedback graphs, aiming to promote the utilization of different feedback patterns in practical applications. Following [Foster and Rakhlin, 2020, Foster et al., 2021, Zhu and Mineiro, 2022], we assume that there is an online regression oracle for supervised learning on the loss. Based on this oracle, we propose SquareCB.G, the first algorithm for contextual bandits with feedback graphs that operates via reduction to regression (Algorithm 1). Eliding regression regret factors, our algorithm achieves the matching optimal regret bounds for deterministic feedback graphs, with $\widetilde{\mathcal{O}}(\sqrt{\alpha T})$ regret for strongly observable graphs and $\widetilde{\mathcal{O}}(d^{\frac{1}{3}}T^{\frac{2}{3}})$ regret for weakly observable graphs, where $\alpha$ and $d$ are respectively the independence number and weakly domination number of the feedback graph (see Section 3.2 for definitions). Notably, SquareCB.G is computationally tractable, requiring the solution to a convex program (Theorem 3.6), which can be readily solved with off-the-shelf convex solvers (Appendix A.3). In addition, we provide closed-form solutions for specific cases of interest (Section 4), leading to a more efficient implementation of our algorithm. Empirical results further showcase the effectiveness of our approach (Section 5).

## 2 Problem Setting and Preliminary

Throughout this paper, we let $[n]$ denote the set $\{1, 2, \ldots, n\}$ for any positive integer $n$. We consider the following contextual bandits problem with informed feedback graphs. The learning process goes in $T$ rounds. At each round $t \in [T]$, an environment selects a context $x_t \in \mathcal{X}$, a (stochastic) directed feedback graph $G_t \in [0, 1]^{\mathcal{A} \times \mathcal{A}}$, and a loss distribution $\mathbb{P}_t : \mathcal{X} \to \Delta([-1, 1]^{\mathcal{A}})$; where $\mathcal{A}$ is the action set with finite cardinality $K$. For convenience, we use $\mathcal{A}$ and $[K]$ interchangeably for denoting the action set. Both $G_t$ and $x_t$ are revealed to the learner at the beginning of each round $t$. Then the learner selects one of the actions $a_t \in \mathcal{A}$, while at the same time, the environment samples a loss vector $\ell_t \in [-1, 1]^{\mathcal{A}}$ from $\mathbb{P}_t(\cdot | x_t)$. The learner then observes some information about $\ell_t$ according to the feedback graph $G_t$. Specifically, for each action $j$, she observes the loss of action $j$ with probability $G_t(a_t, j)$, resulting in a realization $A_t$, which is the set of actions whose loss is observed. With a slight abuse of notation, denote $G_t(\cdot | a)$ as the distribution of $A_t$ when action $a$ is picked. We allow the context $x_t$, the (stochastic) feedback graphs $G_t$ and the loss distribution $\mathbb{P}_t(\cdot | x_t)$ to be selected by an adaptive adversary. When convenient, we will consider $G$ to be a $K$-by-$K$ matrix and utilize matrix notation.

**Other Notations.** Let $\Delta(K)$ denote the set of all Radon probability measures over a set $[K]$. $\mathrm{conv}(S)$ represents the convex hull of a set $S$. Denote $I$ as the identity matrix with an appropriate dimension. For a $K$-dimensional vector $v$, $\mathrm{diag}(v)$ denotes the $K$-by-$K$ matrix with the $i$-th diagonal entry $v_i$ and other entries $0$. We use $\mathbb{R}_{\geq 0}^K$ to denote the set of $K$-dimensional vectors with non-negative entries. For a positive definite matrix $M \in \mathbb{R}^{K \times K}$, we define norm $\|z\|_M = \sqrt{z^\top M z}$ for any vector $z \in \mathbb{R}^K$. We use the $\widetilde{\mathcal{O}}(\cdot)$ notation to hide factors that are polylogarithmic in $K$ and $T$.

**Realizability.** We assume that the learner has access to a known function class $\mathcal{F} \subset (\mathcal{X} \times \mathcal{A} \mapsto [-1, 1])$ which characterizes the mean of the loss for a given context-action pair, and we make the following standard realizability assumption studied in the contextual bandit literature [Agarwal et al., 2012, Foster et al., 2018, Foster and Rakhlin, 2020, Simchi-Levi and Xu, 2021].

**Assumption 1** (Realizability). *There exists a regression function $f^\star \in \mathcal{F}$ such that $\mathbb{E}[\ell_{t,a} \mid x_t] = f^\star(x_t, a)$ for any $a \in \mathcal{A}$ and across all $t \in [T]$.*

Two comments are in order. First, we remark that, similar to [Foster et al., 2020], misspecification can be incorporated while maintaining computational efficiency, but we do not complicate the exposition here. Second, Assumption 1 induces a "semi-adversarial" setting, wherein nature is completely free to determine the context and graph sequences; and has considerable latitude in determining the loss distribution subject to a mean constraint.

**Regret.** For each regression function $f \in \mathcal{F}$, let $\pi_f(x_t) := \operatorname{argmin}_{a \in \mathcal{A}} f(x_t, a)$ denote the induced policy, which chooses the action with the least loss with respective to $f$. Define $\pi^\star := \pi_{f^\star}$ as the optimal policy. We measure the performance of the learner via regret to $\pi^\star$: $\mathbf{Reg}_{\mathsf{CB}} := \sum_{t=1}^{T} \ell_{t,a_t} - \sum_{t=1}^{T} \ell_{t,\pi^\star(x_t)}$, which is the difference between the loss suffered by the learner and the one if the learner applies policy $\pi^\star$.

**Regression Oracle** We assume access to an online regression oracle $\mathbf{Alg}_{\mathsf{Sq}}$ for function class $\mathcal{F}$, which is an algorithm for online learning with squared loss. We consider the following protocol. At each round $t \in [T]$, the algorithm produces an estimator $\widehat{f}_t \in \operatorname{conv}(\mathcal{F})$, then receives a set of context-action-loss tuples $\{(x_t, a, \ell_{t,a})\}_{a \in A_t}$ where $A_t \subseteq \mathcal{A}$. The goal of the oracle is to accurately predict the loss as a function of the context and action, and we evaluate its performance via the square loss $\sum_{a \in A_t} (\widehat{f}_t(x_t, a) - \ell_{t,a})^2$. We measure the oracle's cumulative performance via the following square-loss regret to the best function in $\mathcal{F}$.

**Assumption 2** (Bounded square-loss regret). *The regression oracle* $\mathbf{Alg}_{\mathsf{Sq}}$ *guarantees that for any (potentially adaptively chosen) sequence* $\{(x_t, a, \ell_{t,a})\}_{a \in A_t, t \in [T]}$ *in which* $A_t \subseteq \mathcal{A}$,

$$\sum_{t=1}^{T} \sum_{a \in A_t} \left(\widehat{f}_t(x_t, a) - \ell_{t,a}\right)^2 - \inf_{f \in \mathcal{F}} \sum_{t=1}^{T} \sum_{a \in A_t} \left(f(x_t, a) - \ell_{t,a}\right)^2 \leq \mathbf{Reg}_{\mathsf{Sq}}.$$

For finite $\mathcal{F}$, Vovk's aggregation algorithm yields $\mathbf{Reg}_{\mathsf{Sq}} = \mathcal{O}(\log|\mathcal{F}|)$ [Vovk, 1995]. This regret is dependent upon the scale of the loss function, but this need not be linear with the size of $A_t$, e.g., the loss scale can be bounded by 2 in classification problems. See Foster and Krishnamurthy [2021] for additional examples of online regression algorithms.

## 3 Algorithms and Regret Bounds

In this section, we provide our main algorithms and results.

### 3.1 Algorithms via Minimax Reduction Design

Our approach is to adapt the minimax formulation of [Foster et al., 2021] to contextual bandits with feedback graphs. In the standard contextual bandits setting (that is, $G_t = I$ for all $t$), Foster et al. [2021] define the *Decision-Estimation Coefficient* (DEC) for a parameter $\gamma > 0$ as $\mathsf{dec}_\gamma(\mathcal{F}) := \sup_{\widehat{f} \in \operatorname{conv}(\mathcal{F}), x \in \mathcal{X}} \mathsf{dec}_\gamma(\mathcal{F}; \widehat{f}, x)$, where

$$\begin{aligned} \mathsf{dec}_\gamma(\mathcal{F}; \widehat{f}, x) &:= \inf_{p \in \Delta(K)} \mathsf{dec}_\gamma(p, \mathcal{F}; \widehat{f}, x) \\ &:= \inf_{p \in \Delta(K)} \sup_{\substack{a^\star \in [K] \\ f^\star \in \mathcal{F}}} \mathbb{E}_{a \sim p}\left[f^\star(x, a) - f^\star(x, a^\star) - \frac{\gamma}{4} \cdot \left(\widehat{f}(x, a) - f^\star(x, a)\right)^2\right]. \end{aligned} \quad (1)$$

Their proposed algorithm is as follows. At each round $t$, after receiving the context $x_t$, the algorithm first computes $\widehat{f}_t$ by calling the regression oracle. Then, it solves the solution $p_t$ of the minimax problem defined in Eq. (1) with $\widehat{f}$ and $x$ replaced by $\widehat{f}_t$ and $x_t$. Finally, the algorithm samples an action $a_t$ from the distribution $p_t$ and feeds the observation $(x_t, a_t, \ell_{t,a_t})$ to the oracle. Foster et al. [2021] show that for any value $\gamma$, the algorithm above guarantees that

$$\mathbb{E}[\mathbf{Reg}_{\mathsf{CB}}] \leq T \cdot \mathsf{dec}_\gamma(\mathcal{F}) + \frac{\gamma}{4} \cdot \mathbf{Reg}_{\mathsf{Sq}}. \quad (2)$$

However, the minimax problem Eq. (1) may not be solved efficiently in many cases. Therefore, instead of obtaining the distribution $p_t$ which has the exact minimax value of Eq. (1), Foster et al.

**Algorithm 1** SquareCB.G. Note Theorem 3.6 provides an efficient implementation of Eq. (4).

---

Input: parameter $\gamma \geq 4$, a regression oracle $\mathbf{Alg}_{\mathsf{Sq}}$

**for** $t = 1, 2, \ldots, T$ **do**

> Receive context $x_t$ and directed feedback graph $G_t$.
>
> Obtain an estimator $\widehat{f}_t$ from the oracle $\mathbf{Alg}_{\mathsf{Sq}}$.
>
> Compute the distribution $p_t \in \Delta(K)$ such that $p_t = \operatorname{argmin}_{p \in \Delta(K)} \overline{\mathsf{dec}}_\gamma(p; \widehat{f}_t, x_t, G_t)$, where
>
> $$\overline{\mathsf{dec}}_\gamma(p; \widehat{f}_t, x_t, G_t)$$
> $$:= \sup_{\substack{a^\star \in [K] \\ f^\star \in \Phi}} \mathbb{E}_{a \sim p}\left[ f^\star(x_t, a) - f^\star(x_t, a^\star) - \frac{\gamma}{4} \mathbb{E}_{A \sim G_t(\cdot|a)}\left[ \sum_{a' \in A} (\widehat{f}_t(x_t, a') - f^\star(x_t, a'))^2 \right] \right], \quad (4)$$
>
> and $\Phi := \mathcal{X} \times [K] \mapsto \mathbb{R}$.
>
> Sample $a_t$ from $p_t$ and observe $\{\ell_{t,j}\}_{j \in A_t}$ where $A_t \sim G_t(\cdot|a_t)$.
>
> Feed the tuples $\{(x_t, j, \ell_{t,j})\}_{j \in A_t}$ to the oracle $\mathbf{Alg}_{\mathsf{Sq}}$.

**end**

---

[2021] show that any distribution that gives an upper bound $C_\gamma$ on $\mathsf{dec}_\gamma(p, \mathcal{F}; \widehat{f}, x)$ also works and enjoys a regret bound with $\mathsf{dec}_\gamma(\mathcal{F})$ replaced by $C_\gamma$ in Eq. (2).

To extend this framework to the setting with feedback graph $G$, we define $\mathsf{dec}_\gamma(\mathcal{F}; \widehat{f}, x, G)$ as follows

$$\mathsf{dec}_\gamma(\mathcal{F}; \widehat{f}, x, G)$$
$$:= \inf_{p \in \Delta(K)} \mathsf{dec}_\gamma(p, \mathcal{F}; \widehat{f}, x, G)$$
$$:= \inf_{p \in \Delta(K)} \sup_{\substack{a^\star \in [K] \\ f^\star \in \mathcal{F}}} \mathbb{E}_{a \sim p}\left[ f^\star(x, a) - f^\star(x, a^\star) - \frac{\gamma}{4} \mathbb{E}_{A \sim G(\cdot|a)}\left[ \sum_{a' \in A} (\widehat{f}(x, a') - f^\star(x, a'))^2 \right] \right]. \quad (3)$$

Compared with Eq. (1), the difference is that we replace the squared estimation error on action $a$ by the expected one on the observed set $A \sim G(\cdot|a)$, which intuitively utilizes more feedbacks from the graph structure. When the feedback graph is the identity matrix, we recover Eq. (1). Based on $\mathsf{dec}_\gamma(\mathcal{F}; \widehat{f}, x, G)$, our algorithm SquareCB.G is shown in Algorithm 1. As what is done in [Foster et al., 2021], in order to derive an efficient algorithm, instead of solving the distribution $p_t$ with respect to the supremum over $f^\star \in \mathcal{F}$, we solve $p_t$ that minimize $\overline{\mathsf{dec}}_\gamma(p; \widehat{f}, x_t, G_t)$ (Eq. (4)), which takes supremum over $f^\star \in (\mathcal{X} \times [K] \mapsto \mathbb{R})$, leading to an upper bound on $\mathsf{dec}_\gamma(\mathcal{F}; \widehat{f}, x_t, G_t)$. Then, we receive the loss $\{\ell_{t,j}\}_{j \in A_t}$ and feed the tuples $\{(x_t, j, \ell_{t,j})\}_{j \in A_t}$ to the regression oracle $\mathbf{Alg}_{\mathsf{Sq}}$. Following a similar analysis to [Foster et al., 2021], we show that to bound the regret $\mathbf{Reg}_{\mathsf{CB}}$, we only need to bound $\overline{\mathsf{dec}}_\gamma(p_t; \widehat{f}_t, x_t, G_t)$.

**Theorem 3.1.** *Suppose* $\overline{\mathsf{dec}}_\gamma(p_t; \widehat{f}_t, x_t, G_t) \leq C\gamma^{-\beta}$ *for all* $t \in [T]$ *and some* $\beta > 0$, *Algorithm 1 with* $\gamma = \max\{4, (CT)^{\frac{1}{\beta+1}} \mathbf{Reg}_{\mathsf{Sq}}^{-\frac{1}{\beta+1}}\}$ *guarantees that* $\mathbb{E}\left[\mathbf{Reg}_{\mathsf{CB}}\right] \leq \mathcal{O}\left( C^{\frac{1}{\beta+1}} T^{\frac{1}{\beta+1}} \mathbf{Reg}_{\mathsf{Sq}}^{\frac{\beta}{\beta+1}} \right)$.

The proof is deferred to Appendix A. In Section 3.3, we give an efficient implementation for solving Eq. (4) via reduction to convex programming.

## 3.2 Regret Bounds

In this section, we derive regret bounds for Algorithm 1 when $G_t$'s are specialized to deterministic graphs, i.e., $G_t \in \{0, 1\}^{\mathcal{A} \times \mathcal{A}}$. We utilize discrete graph notation $G = ([K], E)$, where $E \subseteq [K] \times [K]$; and define $N^{\mathrm{in}}(G, j) \triangleq \{i \in \mathcal{A} : (i, j) \in E\}$ as the set of nodes that can observe node $j$. In this case, at each round $t$, the observed node set $A_t$ is a deterministic set which contains any node $i$ satisfying $a_t \in N^{\mathrm{in}}(G_t, i)$. In the following, we introduce several graph-theoretic concepts for deterministic feedback graphs [Alon et al., 2015].

**Strongly/Weakly Observable Graphs.** For a directed graph $G = ([K], E)$, a node $i$ is observable if $N^{\text{in}}(G, i) \neq \emptyset$. An observable node is strongly observable if either $i \in N^{\text{in}}(G, i)$ or $N^{\text{in}}(G, i) = [K]\backslash\{i\}$, and weakly observable otherwise. Similarly, a graph is observable if all its nodes are observable. An observable graph is strongly observable if all nodes are strongly observable, and weakly observable otherwise. Self-aware graphs are a special type of strongly observable graphs where $i \in N^{\text{in}}(G, i)$ for all $i \in [K]$.

**Independent Set and Weakly Dominating Set.** An independence set of a directed graph is a subset of nodes in which no two distinct nodes are connected. The size of the largest independence set of a graph is called its independence number. For a weakly observable graph $G = ([K], E)$, a weakly dominating set is a subset of nodes $D \subseteq [K]$ such that for any node $j$ in $G$ without a self-loop, there exists $i \in D$ such that directed edge $(i, j) \in E$. The size of the smallest weakly dominating set of a graph is called its weak domination number. Alon et al. [2015] show that in non-contextual bandits with a fixed feedback graph $G$, the minimax regret bound is $\widetilde{\Theta}(\sqrt{\alpha T})$ when $G$ is strongly observable and $\widetilde{\Theta}(d^{\frac{1}{3}} T^{\frac{2}{3}})$ when $G$ is weakly observable, where $\alpha$ and $d$ are the independence number and the weak domination number of $G$, respectively.

### 3.2.1 Strongly Observable Graphs

In the following theorem, we show the regret bound of Algorithm 1 for strongly observable graphs.

**Theorem 3.2** (Strongly observable graphs). *Suppose that the feedback graph $G_t$ is deterministic and strongly observable with independence number no more than $\alpha$. Then Algorithm 1 guarantees that*

$$\overline{\text{dec}}_\gamma(p_t; \widehat{f}_t, x_t, G_t) \leq \mathcal{O}\left(\frac{\alpha \log(K\gamma)}{\gamma}\right).$$

In contrast to existing works that derive a closed-form solution of $p_t$ in order to show how large the DEC can be [Foster and Rakhlin, 2020, Foster and Krishnamurthy, 2021], in our case we prove the upper bound of $\overline{\text{dec}}_\gamma(p_t; \widehat{f}_t, x_t, G_t)$ by using the Sion's minimax theorem and the graph-theoretic lemma proven in [Alon et al., 2015]. The proof is deferred to Appendix A.1. Combining Theorem 3.2 and Theorem 3.1, we directly have the following corollary:

**Corollary 3.3.** *Suppose that $G_t$ is deterministic, strongly observable, and has independence number no more than $\alpha$ for all $t \in [T]$. Algorithm 1 with choice $\gamma = \max\left\{4, \sqrt{\alpha T/\mathbf{Reg}_{\text{Sq}}}\right\}$ guarantees that*

$$\mathbb{E}[\mathbf{Reg}_{\text{CB}}] \leq \widetilde{\mathcal{O}}\left(\sqrt{\alpha T \mathbf{Reg}_{\text{Sq}}}\right).$$

For conciseness, we show in Corollary 3.3 that the regret guarantee for Algorithm 1 depends on the largest independence number of $G_t$ over $t \in [T]$. However, we in fact are able to achieve a move adaptive regret bound of order $\widetilde{\mathcal{O}}\left(\sqrt{\sum_{t=1}^T \alpha_t \mathbf{Reg}_{\text{Sq}}}\right)$ where $\alpha_t$ is the independence number of $G_t$.

It is straightforward to achieve this by applying a standard doubling trick on the quantity $\sum_{t=1}^T \alpha_t$, assuming we can compute $\alpha_t$ given $G_t$, but we take one step further and show that it is in fact unnecessary to compute $\alpha_t$ (which, after all, is NP-hard [Karp, 1972]): we provide an adaptive tuning strategy for $\gamma$ by keeping track the the cumulative value of the quantity $\min_{p \in \Delta(K)} \overline{\text{dec}}_\gamma(p; \widehat{f}_t, x_t, G_t)$ and show that this efficient method also achieves the adaptive $\widetilde{\mathcal{O}}\left(\sqrt{\sum_{t=1}^t \alpha_t \mathbf{Reg}_{\text{Sq}}}\right)$ regret guarantee; see Appendix D for details.

### 3.2.2 Weakly Observable Graphs

For the weakly observable graph, we have the following theorem.

**Theorem 3.4** (Weakly observable graphs). *Suppose that the feedback graph $G_t$ is deterministic and weakly observable with weak domination number no more than $d$. Then Algorithm 1 with $\gamma \geq 16d$ guarantees that*

$$\overline{\text{dec}}_\gamma(p_t; \widehat{f}_t, x_t, G_t) \leq \mathcal{O}\left(\sqrt{\frac{d}{\gamma}} + \frac{\widetilde{\alpha} \log(K\gamma)}{\gamma}\right),$$

where $\widetilde{\alpha}$ is the independence number of the subgraph induced by nodes with self-loops in $G_t$.

The proof is deferred to Appendix A.2. Similar to Theorem 3.2, we do not derive a closed-form solution to the strategy $p_t$ but prove this upper bound using the minimax theorem. Combining Theorem 3.4 and Theorem 3.1, we are able to obtain the following regret bound for weakly observable graphs, whose proof is deferred to Appendix A.2.

**Corollary 3.5.** *Suppose that $G_t$ is deterministic, weakly observable, and has weak domination number no more than $d$ for all $t \in [T]$. In addition, suppose that the independence number of the subgraph induced by nodes with self-loops in $G_t$ is no more than $\widetilde{\alpha}$ for all $t \in [T]$. Then, Algorithm 1 with $\gamma = \max\{16d, \sqrt{\widetilde{\alpha}T/\mathbf{Reg_{Sq}}}, d^{\frac{1}{3}}T^{\frac{2}{3}}\mathbf{Reg_{Sq}}^{-\frac{2}{3}}\}$ guarantees that*

$$\mathbb{E}[\mathbf{Reg_{CB}}] \leq \widetilde{\mathcal{O}}\left(d^{\frac{1}{3}}T^{\frac{2}{3}}\mathbf{Reg_{Sq}}^{\frac{1}{3}} + \sqrt{\widetilde{\alpha}T\mathbf{Reg_{Sq}}}\right).$$

Similarly to the strongly observable graph case, we also derive an adaptive tuning strategy for $\gamma$ to achieve a more refined regret bound $\widetilde{\mathcal{O}}\left(\sqrt{\sum_{t=1}^{T}\widetilde{\alpha}_t\mathbf{Reg_{Sq}}} + \left(\sum_{t=1}^{T}\sqrt{d_t}\right)^{\frac{2}{3}}\mathbf{Reg_{Sq}}^{\frac{1}{3}}\right)$ where $\widetilde{\alpha}_t$ is the independence number of the subgraph induced by nodes with self-loops in $G_t$ and $d_t$ is the weakly domination number of $G_t$. This is again achieved *without* explicitly computing $\widetilde{\alpha}_t$ and $d_t$; see Appendix D for details.

### 3.3 Implementation

In this section, we show that solving $\operatorname{argmin}_{p\in\Delta(K)}\overline{\mathsf{dec}}_\gamma(p; \widehat{f}, x, G)$ in Algorithm 1 is equivalent to solving a convex program, which can be easily and efficiently implemented in practice.

**Theorem 3.6.** *Solving $\operatorname{argmin}_{p\in\Delta(K)}\overline{\mathsf{dec}}_\gamma(p; \widehat{f}, x, G)$ is equivalent to solving the following convex optimization problem.*

$$
\begin{aligned}
\min_{p\in\Delta(K), z} \quad & p^\top\widehat{f} + z && (5)\\
\text{subject to} \quad & \forall a \in [K]: \frac{1}{\gamma}\|p - e_a\|^2_{\mathrm{diag}(G^\top p)^{-1}} \leq \widehat{f}(x, a) + z,\\
& G^\top p \succ 0,
\end{aligned}
$$

*where $\widehat{f}$ in the objective is a shorthand for $\widehat{f}(x, \cdot) \in \mathbb{R}^K$, $e_a$ is the $a$-th standard basis vector, and $\succ$ means element-wise greater.*

The proof is deferred to Appendix A.4. Note that this implementation is not restricted to the deterministic feedback graphs but applies to the general stochastic feedback graph case. In Appendix A.3, we provide the 20 lines of Python code that solves Eq. (5).

## 4 Examples with Closed-Form Solutions

In this section, we present examples and corresponding closed-form solutions of $p$ that make the value $\overline{\mathsf{dec}}_\gamma(p; \widehat{f}, x, G)$ upper bounded by at most a constant factor of $\min_p \overline{\mathsf{dec}}_\gamma(p; \widehat{f}, x, G)$. This offers an alternative to solving the convex program defined in Theorem 3.6 for special (and practically relevant) cases, thereby enhancing the efficiency of our algorithm. All the proofs are deferred to Appendix B.

**Cops-and-Robbers Graph.** The "cops-and-robbers" feedback graph $G_{\mathrm{CR}} = 11^\top - I$, also known as the loopless clique, is the full feedback graph removing self-loops. Therefore, $G_{\mathrm{CR}}$ is strongly observable with independence number $\alpha = 1$. Let $a_1$ be the node with the smallest value of $\widehat{f}$ and $a_2$ be the node with the second smallest value of $\widehat{f}$. Our proposed closed-form distribution $p$ is only supported on $\{a_1, a_2\}$ and defined as follows:

$$p_{a_1} = 1 - \frac{1}{2 + \gamma(\widehat{f}_{a_2} - \widehat{f}_{a_1})}, \quad p_{a_2} = \frac{1}{2 + \gamma(\widehat{f}_{a_2} - \widehat{f}_{a_1})}. \tag{6}$$

In the following proposition, we show that with the construction of $p$ in Eq. (6), $\overline{\mathsf{dec}}_\gamma(p; \widehat{f}, x, G_{\mathrm{CR}})$ is upper bounded by $\mathcal{O}(1/\gamma)$, which matches the order of $\min_p \overline{\mathsf{dec}}_\gamma(p; \widehat{f}, x, G)$ based on Theorem 3.2 since $\alpha = 1$.

**Proposition 1.** *When $G = G_{\mathrm{CR}}$, given any $\widehat{f}$, context $x$, the closed-form distribution $p$ in Eq. (6) guarantees that $\overline{\mathsf{dec}}_\gamma(p; \widehat{f}, x, G_{\mathrm{CR}}) \leq \mathcal{O}\left(\frac{1}{\gamma}\right)$.*

**Apple Tasting Graph.** The apple tasting feedback graph $G_{\mathrm{AT}} = \begin{bmatrix} 1 & 1 \\ 0 & 0 \end{bmatrix}$ consists of two nodes, where the first node reveals all and the second node reveals nothing. This scenario was originally proposed by Helmbold et al. [2000] and recently denoted the spam filtering graph [van der Hoeven et al., 2021]. The independence number of $G_{\mathrm{AT}}$ is 1. Let $\widehat{f}_1$ be the oracle prediction for the first node and let $\widehat{f}_2$ be the prediction for the second node. We present a closed-form solution $p$ for Eq. (4) as follows:

$$p_1 = \begin{cases} 1 & \widehat{f}_1 \leq \widehat{f}_2 \\ \frac{2}{4 + \gamma(\widehat{f}_1 - \widehat{f}_2)} & \widehat{f}_1 > \widehat{f}_2 \end{cases}, \qquad p_2 = 1 - p_1. \tag{7}$$

We show that this distribution $p$ satisfies that $\overline{\mathsf{dec}}_\gamma(p; \widehat{f}, x, G_{\mathrm{AT}})$ is upper bounded by $\mathcal{O}(1/\gamma)$ in the following proposition. We remark that directly applying results from [Foster et al., 2021] cannot lead to a valid upper bound since the second node does not have a self-loop.

**Proposition 2.** *When $G = G_{\mathrm{AT}}$, given any $\widehat{f}$, context $x$, the closed-form distribution $p$ in Eq. (7) guarantees that $\overline{\mathsf{dec}}_\gamma(p; \widehat{f}, x, G_{\mathrm{AT}}) \leq \mathcal{O}(\frac{1}{\gamma})$.*

**Inventory Graph.** In this application, the algorithm needs to decide the inventory level in order to fulfill the realized demand arriving at each round. Specifically, there are $K$ possible chosen inventory levels $a_1 < a_2 < \ldots < a_K$ and the feedback graph $G_{\mathrm{inv}}$ has entries $G(i, j) = 1$ for all $1 \leq j \leq i \leq K$ and $G(i, j) = 0$ otherwise, meaning that picking the inventory level $a_i$ informs about all actions $a_{j \leq i}$. This is because items are consumed until either the demand or the inventory is exhausted. The independence number of $G_{\mathrm{inv}}$ is 1. Therefore, (very) large $K$ is statistically tractable, but naively solving the convex program Eq. (5) requires superlinear in $K$ computational cost. We show in the following proposition that there exists an analytic form of $p$, which guarantees that $\overline{\mathsf{dec}}_\gamma(p; \widehat{f}, x, G_{\mathrm{inv}})$ can be bounded by $\mathcal{O}(1/\gamma)$.

**Proposition 3.** *When $G = G_{\mathrm{inv}}$, given any $\widehat{f}$, context $x$, there exists a closed-form distribution $p \in \Delta(K)$ guaranteeing that $\overline{\mathsf{dec}}_\gamma(p; \widehat{f}, x, G_{\mathrm{inv}}) \leq \mathcal{O}(\frac{1}{\gamma})$, where $p$ is defined as follows: $p_j = \max\{\frac{1}{1 + \gamma(\widehat{f}_j - \min_i \widehat{f}_i)} - \sum_{j' > j} p_{j'}, 0\}$ for all $j \in [K]$.*

**Undirected Self-Aware Graph.** For the undirected and self-aware feedback graph $G$, which means that $G$ is symmetric and has diagonal entries all 1, we also show that a certain closed-form solution of $p$ satisfies that $\overline{\mathsf{dec}}_\gamma(p; \widehat{f}, x, G)$ is bounded by $\mathcal{O}(\frac{\alpha}{\gamma})$.

**Proposition 4.** *When $G$ is an undirected self-aware graph, given any $\widehat{f}$, context $x$, there exists a closed-form distribution $p \in \Delta(K)$ guaranteeing that $\overline{\mathsf{dec}}_\gamma(p; \widehat{f}, x, G) \leq \mathcal{O}\left(\frac{\alpha}{\gamma}\right)$.*

## 5 Experiments

In this section, we use empirical results to demonstrate the significant benefits of SquareCB.G in leveraging the graph feedback structure and its superior effectiveness compared to SquareCB. Following Foster and Krishnamurthy [2021], we use progressive validation (PV) loss as the evaluation metric, defined as $L_{\mathrm{pv}}(T) = \frac{1}{T} \sum_{t=1}^{T} \ell_{t, a_t}$. All the feedback graphs used in the experiments are deterministic. We run experiments on CPU Intel Xeon Gold 6240R 2.4G and the convex program solver is implemented via Vowpal Wabbit [Langford et al., 2007].

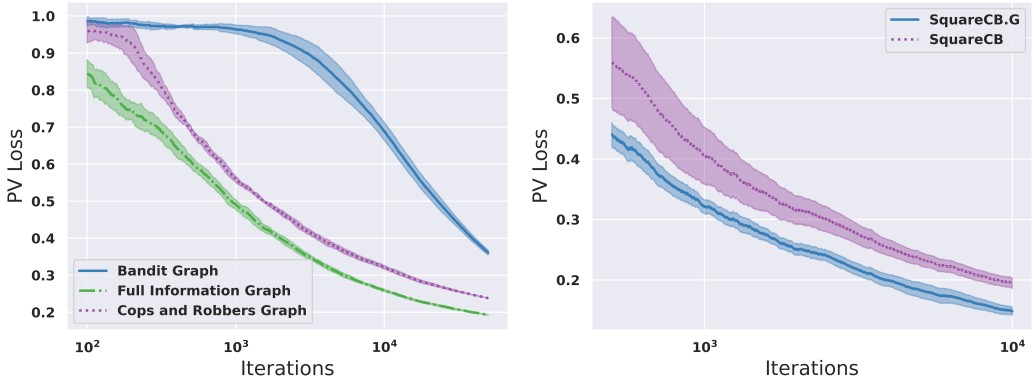

Figure 1: **Left figure**: Performance of SquareCB.G on RCV1 dataset under three different feedback graphs. **Right figure**: Performance comparison between SquareCB.G and SquareCB under random directed self-aware feedback graphs.

### 5.1  SquareCB.G **under Different Feedback Graphs**

In this subsection, we show that our SquareCB.G benefits from considering the graph structure by evaluating the performance of SquareCB.G under three different feedback graphs. We conduct experiments on RCV1 dataset and leave the implementation details in Appendix C.1.

The performances of SquareCB.G under bandit graph, full information graph and cops-and-robbers graph are shown in the left part of Figure 1. We observe that SquareCB.G performs the best under full information graph and performs worst under bandit graph. Under the cops-and-robbers graph, much of the gap between bandit and full information is eliminated. This improvement demonstrates the benefit of utilizing graph feedback for accelerating learning.

### 5.2  **Comparison between** SquareCB.G **and** SquareCB

In this subsection, we compare the effectiveness of SquareCB.G with the SquareCB algorithm. To ensure a fair comparison, both algorithms update the regressor using the same feedbacks based on the graph. The only distinction lies in how they calculate the action probability distribution. We summarize the main results here and leave the implementation details in Appendix C.2.

#### 5.2.1  **Results on Random Directed Self-aware Graphs**

We conduct experiments on RCV1 dataset using random directed self-aware feedback graphs. Specifically, at round $t$, the deterministic feedback graph $G_t$ is generated as follows. The diagonal elements of $G_t$ are all 1, and each off-diagonal entry is drawn from a Bernoulli($3/4$) distribution. The results are presented in the right part of Figure 1. Our SquareCB.G consistently outperforms SquareCB and demonstrates lower variance, particularly when the number of iterations was small. This is because when there are fewer samples available to train the regressor, it is more crucial to design an effective algorithm that can leverage the graph feedback information.

#### 5.2.2  **Results on Synthetic Inventory Dataset**

In the inventory graph experiments, we create a synthetic inventory dataset and design a loss function for each inventory level $a_t \in [0, 1]$ with demand $d_t \in [0, 1]$. Since the action set $[0, 1]$ is continuous, we discretize the action set in two different ways to apply the algorithms.

**Fixed discretized action set.**  In this setting, we discretize the action set using fixed grid size $\varepsilon \in \{\frac{1}{100}, \frac{1}{300}, \frac{1}{500}\}$, which leads to a finite action set $\mathcal{A}$ of size $\frac{1}{\varepsilon} + 1$. Note that according to Theorem 3.2, our regret *does not* scale with the size of the action set (to within polylog factors), as the independence number is always 1. The results are shown in the left part of Figure 2.

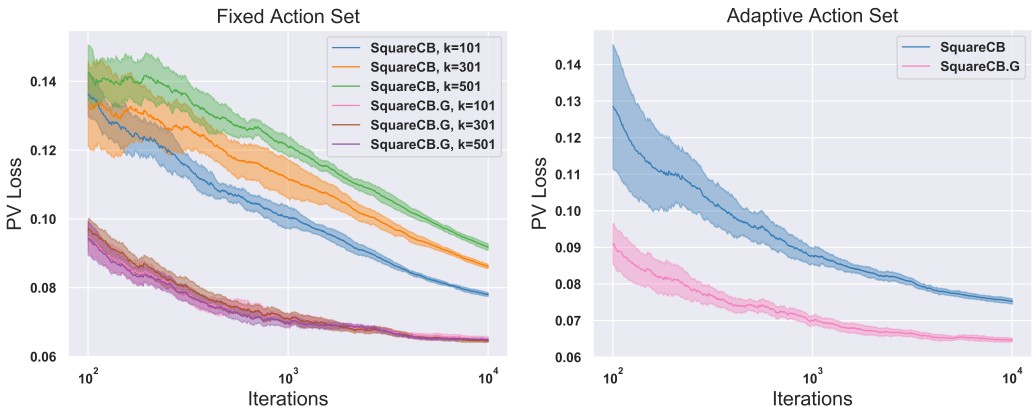

Figure 2: Performance comparison between SquareCB.G and SquareCB on synthetic inventory dataset. **Left figure**: Results under fixed discretized action set. **Right figure**: Results under adaptive discretization of the action set. Both figures show the superiority of SquareCB.G compared with SquareCB.

We remark several observations from the results. First, our algorithm SquareCB.G outperforms SquareCB for all choices $K \in \{101, 301, 501\}$. This indicates that SquareCB.G utilizes a better exploration scheme and effectively leverages the structure of $G_{\mathrm{inv}}$. Second, we observe that SquareCB.G indeed does not scale with the size of the discretized action set $\mathcal{A}$, since under different discretization scales, SquareCB.G has similar performances and the slight differences are from the improved approximation error with finer discretization. This matches the theoretical guarantee that we prove in Theorem 3.2. On the other hand, SquareCB does perform worse when the size of the action set increases, matching its theoretical guarantee which scales with the square root of the size of the action set.

**Adaptively changing action set.** In this setting, we adaptively discretize the action set $[0, 1]$ according to the index of the current round. Specifically, for SquareCB.G, we uniformly discretize the action set $[0, 1]$ with size $\lceil\sqrt{t}\rceil$, whose total discretization error is $\mathcal{O}(\sqrt{T})$ due to the Lipschitzness of the loss function. For SquareCB, to optimally balance the dependency on the size of the action set and the discretization error, we uniformly discretize the action set $[0, 1]$ into $\lceil t^{\frac{1}{3}} \rceil$ actions. The results are illustrated in the right part of Figure 2. We can observe that SquareCB.G consistently outperforms SquareCB by a clear margin.

## 6 Related Work

Multi-armed bandits with feedback graphs have been extensively studied. An early example is the apple tasting problem of Helmbold et al. [2000]. The general formulation was introduced by Mannor and Shamir [2011]. Alon et al. [2015] characterized the minimax rates in terms of graph-theoretic quantities. Follow-on work includes relaxing the assumption that the graph is observed prior to decision [Cohen et al., 2016]; analyzing the distinction between the stochastic and adversarial settings [Alon et al., 2017]; considering stochastic feedback graphs [Li et al., 2020, Esposito et al., 2022]; instance-adaptivity [Ito et al., 2022]; data-dependent regret bound [Lykouris et al., 2018, Lee et al., 2020]; and high-probability regret under adaptive adversary [Neu, 2015, Luo et al., 2023].

The contextual bandit problem with feedback graphs has received relatively less attention. Wang et al. [2021] provide algorithms for adversarial linear bandits with uninformed graphs and stochastic contexts. However, this work assumes several unrealistic assumptions on both the policy class and the context space and is not comparable to our setting, since we consider the informed graph setting with adversarial context. Singh et al. [2020] study a stochastic linear bandits with informed feedback graphs and are able to improve over the instance-optimal regret bound for bandits derived in [Lattimore and Szepesvari, 2017] by utilizing the additional graph-based feedbacks.

Our work is also closely related to the recent progress in designing efficient algorithms for classic contextual bandits. Starting from [Langford and Zhang, 2007], numerous works have been done to the development of practically efficient algorithms, which are based on reduction to either cost-sensitive classification oracles [Dudik et al., 2011, Agarwal et al., 2014] or online regression oracles [Foster and Rakhlin, 2020, Foster et al., 2020, 2021, Zhu and Mineiro, 2022]. Following the latter trend, our work assumes access to an online regression oracle and extends the classic bandit problems to the bandits with general feedback graphs.

# 7 Discussion

In this paper, we consider the design of practical contextual bandits algorithm with provable guarantees. Specifically, we propose the first efficient algorithm that achieves near-optimal regret bound for contextual bandits with general directed feedback graphs with an online regression oracle.

While we study the informed graph feedback setting, where the entire feedback graph is exposed to the algorithm prior to each decision, many practical problems of interest are possibly uninformed graph feedback problems, where the graph is unknown at the decision time. It is unclear how to formulate an analogous minimax problem to Eq. (1) under the uninformed setting. One idea is to consume the additional feedback in the online regressor and adjust the prediction loss to reflect this additional structure, e.g., using the more general version of the E2D framework which incorporates arbitrary side observations [Foster et al., 2021]. Cohen et al. [2016] consider this uninformed setting in the non-contextual case and prove a sharp distinction between the adversarial and stochastic settings: even if the graphs are all strongly observable with bounded independence number, in the adversarial setting the minimax regret is $\Theta(T)$ whereas in the stochastic setting the minimax regret is $\Theta(\sqrt{\alpha T})$. Intriguingly, our setting is semi-adversarial due to realizability of the mean loss, and therefore it is apriori unclear whether the negative adversarial result applies.

In addition, bandits with graph feedback problems often present with associated policy constraints, e.g., for the apple tasting problem, it is natural to rate limit the informative action. Therefore, another interesting direction is to combine our algorithm with the recent progress in contextual bandits with knapsack [Slivkins and Foster, 2022], leading to more practical algorithms.

**Acknowledgments** HL and MZ are supported by NSF Awards IIS-1943607.

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

# A    Omitted Details in Section 3

**Theorem 3.1.** *Suppose* $\overline{\mathsf{dec}}_\gamma(p_t; \widehat{f}_t, x_t, G_t) \leq C\gamma^{-\beta}$ *for all* $t \in [T]$ *and some* $\beta > 0$, *Algorithm 1 with* $\gamma = \max\{4, (CT)^{\frac{1}{\beta+1}} \mathbf{Reg}_{\mathsf{Sq}}^{-\frac{1}{\beta+1}}\}$ *guarantees that* $\mathbb{E}\left[\mathbf{Reg}_{\mathsf{CB}}\right] \leq \mathcal{O}\left(C^{\frac{1}{\beta+1}} T^{\frac{1}{\beta+1}} \mathbf{Reg}_{\mathsf{Sq}}^{\frac{\beta}{\beta+1}}\right)$.

*Proof.* Following [Foster et al., 2020], we decompose $\mathbf{Reg}_{\mathsf{CB}}$ as follows:

$$\mathbb{E}[\mathbf{Reg}_{\mathsf{CB}}]$$

$$= \mathbb{E}\left[\sum_{t=1}^{T} f^\star(x_t, a_t) - \sum_{t=1}^{T} f^\star(x_t, \pi^\star(x_t))\right]$$

$$= \mathbb{E}\left[\sum_{t=1}^{T} \left(f^\star(x_t, a_t) - f^\star(x_t, \pi^\star(x_t)) - \frac{\gamma}{4}\mathbb{E}_{A \sim G_t(\cdot|a_t)}\left[\sum_{a \in A}\left(\widehat{f}_t(x_t, a) - f^\star(x_t, a)\right)^2\right]\right)\right]$$

$$\quad + \frac{\gamma}{4}\mathbb{E}\left[\sum_{t=1}^{T}\mathbb{E}_{A \sim G_t(\cdot|a_t)}\left[\sum_{a \in A}\left(\widehat{f}_t(x_t, a) - f^\star(x_t, a)\right)^2\right]\right]$$

$$\leq \mathbb{E}\left[\sum_{t=1}^{T} \max_{\substack{a^\star \in [K] \\ f \in (\mathcal{X} \times [K] \mapsto \mathbb{R})}} \mathbb{E}_{a_t \sim p_t}\left[f(x_t, a_t) - f(x_t, a^\star) - \frac{\gamma}{4}\mathbb{E}_{A \sim G_t(\cdot|a_t)}\left[\sum_{a \in A}\left(\widehat{f}_t(x_t, a) - f(x_t, a)\right)^2\right]\right]\right]$$

$$\quad + \frac{\gamma}{4}\mathbb{E}\left[\sum_{t=1}^{T}\mathbb{E}_{A \sim G_t(\cdot|a_t)}\left[\sum_{a \in A}\left(\widehat{f}_t(x_t, a) - f^\star(x_t, a)\right)^2\right]\right]$$

$$= \mathbb{E}\left[\sum_{t=1}^{T}\overline{\mathsf{dec}}_\gamma(p_t; \widehat{f}_t, x_t, G_t)\right] + \frac{\gamma}{4}\mathbb{E}\left[\sum_{t=1}^{T}\mathbb{E}_{A \sim G_t(\cdot|a_t)}\left[\sum_{a \in A}\left(\widehat{f}_t(x_t, a) - f^\star(x_t, a)\right)^2\right]\right] \qquad (8)$$

$$\leq CT\gamma^{-\beta} + \frac{\gamma}{4}\mathbb{E}\left[\sum_{t=1}^{T}\mathbb{E}_{A \sim G_t(\cdot|a_t)}\left[\sum_{a \in A}\left(\widehat{f}_t(x_t, a) - f^\star(x_t, a)\right)^2\right]\right].$$

Next, since $\mathbb{E}[\ell_{t,a} \mid x_t] = f^\star(x_t, a)$ for all $t \in [T]$ and $a \in \mathcal{A}$, we know that

$$\mathbb{E}\left[\sum_{t=1}^{T}\mathbb{E}_{A \sim G_t(\cdot|a_t)}\left[\sum_{a \in A}\left(\widehat{f}_t(x_t, a) - f^\star(x_t, a)\right)^2\right]\right]$$

$$= \mathbb{E}\left[\sum_{t=1}^{T}\mathbb{E}_{A \sim G_t(\cdot|a_t)}\left[\sum_{a \in A}\left(\widehat{f}_t(x_t, a) - \ell_{t,a}\right)^2 - \sum_{a \in A}\left(f^\star(x_t, a) - \ell_{t,a}\right)^2\right]\right] \leq \mathbf{Reg}_{\mathsf{Sq}}, \quad (9)$$

where the final inequality is due to Assumption 2.

Therefore, we have

$$\mathbb{E}[\mathbf{Reg}_{\mathsf{CB}}] \leq CT\gamma^{-\beta} + \frac{\gamma}{4}\mathbf{Reg}_{\mathsf{Sq}}.$$

Picking $\gamma = \max\left\{4, \left(\frac{CT}{\mathbf{Reg}_{\mathsf{Sq}}}\right)^{\frac{1}{\beta+1}}\right\}$, we obtain that

$$\mathbb{E}\left[\mathbf{Reg}_{\mathsf{CB}}\right] \leq \mathcal{O}\left(C^{\frac{1}{\beta+1}} T^{\frac{1}{\beta+1}} \mathbf{Reg}_{\mathsf{Sq}}^{\frac{\beta}{\beta+1}}\right).$$

$\square$

## A.1    Proof of Theorem 3.2

Before proving Theorem 3.2, we first show the following key lemma, which is useful in proving that $\overline{\mathsf{dec}}_\gamma(p; \widehat{f}, x, G)$ is convex for both strongly and weakly observable feedback graphs $G$. We

highlight that the convexity of $\overline{\mathsf{dec}}_\gamma(p; \widehat{f}, x, G)$ is crucial for both proving the upper bound of $\min_{p \in \Delta(K)} \overline{\mathsf{dec}}_\gamma(p; \widehat{f}, x, G)$ and showing the efficiency of Algorithm 1.

**Lemma A.1.** *Suppose $u, v, x \in \mathbb{R}^d$ with $\langle v, x \rangle > 0$. Then both $g(x) = \frac{\langle u, x \rangle^2}{\langle v, x \rangle}$ and $h(x) = \frac{(1 - \langle u, x \rangle)^2}{\langle v, x \rangle}$ are convex in $x$.*

*Proof.* The function $f(x, y) = x^2/y$ is convex for $y > 0$ due to

$$\nabla^2 f(x, y) = \frac{2}{y^3} \begin{bmatrix} y \\ -x \end{bmatrix} \begin{bmatrix} y \\ -x \end{bmatrix}^\top \succeq 0.$$

By composition with affine functions, both $g(x) = f(\langle u, x \rangle, \langle v, x \rangle)$ and $h(x) = f(1 - \langle u, x \rangle, \langle v, x \rangle)$ are convex. $\square$

**Theorem 3.2** (Strongly observable graphs). *Suppose that the feedback graph $G_t$ is deterministic and strongly observable with independence number no more than $\alpha$. Then Algorithm 1 guarantees that*

$$\overline{\mathsf{dec}}_\gamma(p_t; \widehat{f}_t, x_t, G_t) \leq \mathcal{O}\left( \frac{\alpha \log(K\gamma)}{\gamma} \right).$$

*Proof.* For conciseness, we omit the subscript $t$. Direct calculation shows that for all $a^\star \in [K]$,

$$\mathbb{E}_{a \sim p} \left[ f^\star(x, a) - f^\star(x, a^\star) - \frac{\gamma}{4} \sum_{a' \in N^{\mathrm{in}}(G, a)} (\widehat{f}(x, a') - f^\star(x, a'))^2 \right]$$

$$= \sum_{a=1}^{K} p_a f^\star(x, a) - f^\star(x, a^\star) - \frac{\gamma}{4} \sum_{a=1}^{K} W_a \left( \widehat{f}(x, a) - f^\star(x, a) \right)^2,$$

where $W_a = \sum_{a' \in N^{\mathrm{in}}(G, a)} p_{a'}$. Therefore, taking the gradient over $f^*(x, \cdot)$ and we know that

$$\sup_{f^\star \in (\mathcal{X} \times [K] \mapsto \mathbb{R})} \left[ \sum_{a=1}^{K} p_a f^\star(x, a) - f^\star(x, a^\star) - \frac{\gamma}{4} \sum_{a=1}^{K} W_a \left( \widehat{f}(x, a) - f^\star(x, a) \right)^2 \right]$$

$$= \sum_{a=1}^{K} p_a \widehat{f}(x, a) - \widehat{f}(x, a^\star) + \frac{1}{\gamma} \| p - e_{a^\star} \|^2_{\mathrm{diag}(W)^{-1}}.$$

Then, denote $\widehat{f} \in \mathbb{R}^K$ to be $\widehat{f}(x, \cdot)$ and consider the following minimax form:

$$\inf_{p \in \Delta(K)} \sup_{a^\star \in \mathcal{A}} \left\{ \sum_{a=1}^{K} p_a \widehat{f}(x, a) - \widehat{f}(x, a^\star) + \frac{1}{\gamma} \| p - e_{a^\star} \|^2_{\mathrm{diag}(W)^{-1}} \right\}$$

$$= \min_{p \in \Delta(K)} \max_{a^\star \in \mathcal{A}} \left\{ \sum_{a=1}^{K} p_a \widehat{f}(x, a) - \widehat{f}(x, a^\star) + \frac{1}{\gamma} \sum_{a \neq a^\star} \frac{p_a^2}{W_a} + \frac{1}{\gamma} \frac{(1 - p_{a^\star})^2}{W_{a^\star}} \right\} \quad (10)$$

$$= \min_{p \in \Delta_K} \max_{q \in \Delta_K} \left\{ \sum_{a=1}^{K} (p_a - q_a) \widehat{f}_a + \frac{1}{\gamma} \sum_{a=1}^{K} \frac{p_a^2 (1 - q_a)}{W_a} + \sum_{a=1}^{K} \frac{q_a (1 - p_a)^2}{\gamma W_a} \right\} \quad (11)$$

$$= \max_{q \in \Delta_K} \min_{p \in \Delta_K} \left\{ \sum_{a=1}^{K} (p_a - q_a) \widehat{f}_a + \frac{1}{\gamma} \sum_{a=1}^{K} \frac{p_a^2 (1 - q_a)}{W_a} + \sum_{a=1}^{K} \frac{q_a (1 - p_a)^2}{\gamma W_a} \right\}, \quad (12)$$

where the last equality is due to Sion's minimax theorem and the fact that Eq. (10) is convex in $p \in \Delta(K)$ by applying Lemma A.1 with $u = e_a$ and $v = g_a$ for each $a \in [K]$, where $g_a \in \{0, 1\}^K$ is defined as $g_{a,i} = \mathbb{1}\{(i, a) \in E\}$, $G = ([K], E)$, $\forall i \in [K]$.

Choose $p_a = (1 - \frac{1}{\gamma}) q_a + \frac{1}{\gamma K}$ for all $a \in [K]$. Let $S$ be the set of nodes in $[K]$ that have a self-loop. Then we can upper bound the value above as follows:

$$\max_{q \in \Delta(K)} \min_{p \in \Delta(K)} \left\{ \sum_{a=1}^{K} (p_a - q_a) \widehat{f}_a + \frac{1}{\gamma} \sum_{a=1}^{K} \frac{p_a^2 (1 - q_a)}{W_a} + \sum_{a=1}^{K} \frac{q_a (1 - p_a)^2}{\gamma W_a} \right\}$$

$$\leq \max_{q\in\Delta(K)} \left\{ \frac{2}{\gamma} + \frac{1}{\gamma} \sum_{a=1}^{K} \frac{\left((1-\frac{1}{\gamma})q_a + \frac{1}{\gamma K}\right)^2 (1-q_a) + q_a\left(1 - (1-\frac{1}{\gamma})q_a - \frac{1}{\gamma K}\right)^2}{W_a} \right\}$$

$$\leq \max_{q\in\Delta(K)} \left\{ \frac{2}{\gamma} + \frac{1}{\gamma} \sum_{a=1}^{K} \frac{2\left((1-\frac{1}{\gamma})^2 q_a^2 + \frac{1}{\gamma^2 K^2}\right)(1-q_a) + q_a\left(1 - (1-\frac{1}{\gamma})q_a\right)^2}{W_a} \right\}$$

$$\leq \max_{q\in\Delta(K)} \left\{ \frac{2}{\gamma} + \frac{2}{\gamma^2} + \frac{1}{\gamma} \sum_{a=1}^{K} \frac{2q_a^2(1-q_a) + 2q_a(1-q_a)^2 + \frac{2q_a^3}{\gamma^2}}{W_a} \right\}$$

$$(W_a = \textstyle\sum_{j\in N^{\mathrm{in}}(G,a)} p_j \geq \frac{1}{\gamma K} \text{ for all } a \in [K])$$

$$\leq \max_{q\in\Delta(K)} \left\{ \frac{2}{\gamma} + \frac{2}{\gamma^2} + \frac{2}{\gamma} \sum_{a=1}^{K} \frac{q_a(1-q_a)}{W_a} + \frac{2}{\gamma^3} \sum_{a=1}^{K} \frac{q_a^3}{W_a} \right\}$$

$$= \max_{q\in\Delta(K)} \left\{ \frac{2}{\gamma} + \frac{2}{\gamma^2} + \frac{2}{\gamma} \sum_{a=1}^{K} \frac{q_a(1-q_a)}{W_a} + \frac{2}{\gamma^3} \sum_{a\in S} \frac{q_a^3}{W_a} + \frac{2}{\gamma^3} \sum_{a\notin S} \frac{q_a^3}{W_a} \right\} \tag{13}$$

$$\leq \max_{q\in\Delta(K)} \left\{ \frac{2}{\gamma} + \frac{2}{\gamma^2} + \frac{2}{\gamma} \sum_{a=1}^{K} \frac{q_a(1-q_a)}{W_a} + \frac{2}{\gamma^3} \sum_{a\in S} q_a^2 + \frac{2}{\gamma^3} \sum_{a\notin S} \frac{q_a^3}{\frac{K-1}{\gamma K}} \right\}$$

$$(\text{if } a \notin S, W_a = 1 - p_a \geq \tfrac{K-1}{\gamma K})$$

$$\leq \max_{q\in\Delta(K)} \left\{ \frac{8}{\gamma} + \frac{2}{\gamma} \sum_{a=1}^{K} \frac{q_a(1-q_a)}{W_a} \right\}. \tag{$K \geq 2$}$$

Next we bound $\frac{2q_a(1-q_a)}{W_a}$ for each $a \in [K]$. If $a \in [K]\setminus S$, we have $W_a = 1 - p_a$ and

$$\frac{2q_a(1-q_a)}{W_a} \leq \frac{2q_a(1-q_a)}{1 - (1-\frac{1}{\gamma})q_a - \frac{1}{\gamma K}} \leq \frac{2q_a(1-q_a)}{(1-\frac{1}{\gamma})(1-q_a) + \frac{K-1}{\gamma K}} \leq \frac{2}{1-\frac{1}{\gamma}} q_a \leq 4q_a. \tag{14}$$

If $a \in S$, we know that

$$\sum_{a\in S} \frac{2q_a(1-q_a)}{W_a} \leq \sum_{a\in S} \frac{2q_a(1-q_a)}{\sum_{j\in N^{\mathrm{in}}(G,a)}\left((1-\frac{1}{\gamma})q_j + \frac{1}{\gamma K}\right)}$$

$$\leq \frac{\gamma}{\gamma - 1} \sum_{a\in S} \frac{2\left((1-\frac{1}{\gamma})q_a + \frac{1}{\gamma K}\right)(1-q_a)}{\sum_{j\in N^{\mathrm{in}}(G,a)}\left((1-\frac{1}{\gamma})q_j + \frac{1}{\gamma K}\right)}$$

$$\leq 4 \sum_{a\in S} \frac{\left((1-\frac{1}{\gamma})q_a + \frac{1}{\gamma K}\right)}{\sum_{j\in N^{\mathrm{in}}(G,a)}\left((1-\frac{1}{\gamma})q_j + \frac{1}{\gamma K}\right)} \leq \mathcal{O}(\alpha\log(K\gamma)), \tag{15}$$

where the last inequality is due to Lemma 5 in Alon et al. [2015]. We include this lemma (Lemma E.1) for completeness. Combining all the above inequalities, we obtain that

$$\inf_{p\in\Delta(K)} \sup_{a^\star\in\mathcal{A}} \left\{ \sum_{a=1}^{K} p_a\widehat{f}(x,a) - \widehat{f}(x,a^\star) + \frac{1}{\gamma}\|p - e_{a^\star}\|^2_{\mathrm{diag}(W)^{-1}} \right\}$$

$$= \max_{q\in\Delta(K)} \min_{p\in\Delta(K)} \left\{ \sum_{a=1}^{K} (p_a - q_a)\widehat{f}_a + \frac{1}{\gamma} \sum_{a=1}^{K} \frac{p_a^2(1-q_a)}{W_a} + \sum_{a=1}^{K} \frac{q_a(1-p_a)^2}{\gamma W_a} \right\}$$

$$\leq \max_{q\in\Delta(K)} \left\{ \frac{8}{\gamma} + \frac{2}{\gamma} \sum_{a=1}^{K} \frac{q_a(1-q_a)}{W_a} \right\} \leq \mathcal{O}\left(\frac{\alpha\log(K\gamma)}{\gamma}\right).$$

$$\qquad\square$$

## A.2 Proof of Theorem 3.4

**Theorem 3.4** (Weakly observable graphs). *Suppose that the feedback graph $G_t$ is deterministic and weakly observable with weak domination number no more than $d$. Then Algorithm 1 with $\gamma \geq 16d$ guarantees that*

$$\overline{\mathsf{dec}}_\gamma(p_t; \widehat{f}_t, x_t, G_t) \leq \mathcal{O}\left(\sqrt{\frac{d}{\gamma}} + \frac{\widetilde{\alpha}\log(K\gamma)}{\gamma}\right),$$

*where $\widetilde{\alpha}$ is the independence number of the subgraph induced by nodes with self-loops in $G_t$.*

*Proof.* Similar to the strongly observable graphs setting, for weakly observable graphs, we know that

$$\overline{\mathsf{dec}}_\gamma(p; \widehat{f}, x, G)$$
$$= \max_{q \in \Delta_K} \min_{p \in \Delta_K} \left\{ \sum_{a=1}^{K}(p_a - q_a)\widehat{f}_a + \frac{1}{\gamma}\sum_{a=1}^{K}\frac{p_a^2(1-q_a)}{W_a} + \sum_{a=1}^{K}\frac{q_a(1-p_a)^2}{\gamma W_a} \right\}. \quad (16)$$

Choose $p_a = (1 - \frac{1}{\gamma} - \eta d)q_a + \frac{1}{\gamma K} + \eta\mathbb{1}\{a \in D\}$ where $D$ with $|D| = d$ is the minimum weak dominating set of $G$ and $0 < \eta \leq \frac{1}{4d}$ is some parameter to be chosen later. Substituting the form of $p$ to Eq. (16) and using the fact that $|\widehat{f}_a| \leq 1$ for all $a \in [K]$, we can obtain that

$$\overline{\mathsf{dec}}_\gamma(p; \widehat{f}, x, G)$$
$$\leq \max_{q \in \Delta_K}\left\{ \frac{2}{\gamma} + \eta d + \frac{1}{\gamma}\sum_{a=1}^{K}\frac{p_a^2(1-q_a)}{W_a} + \sum_{a=1}^{K}\frac{q_a(1-p_a)^2}{\gamma W_a} \right\}.$$

Then we can upper bound the value above as follows:

$$\overline{\mathsf{dec}}_\gamma(p; \widehat{f}, x, G)$$
$$\leq \max_{q \in \Delta_K}\left\{ \frac{2}{\gamma} + \eta d + \frac{1}{\gamma}\sum_{a=1}^{K}\frac{\left((1 - \frac{1}{\gamma} - \eta d)q_a + \frac{1}{\gamma K} + \eta\mathbb{1}\{a \in D\}\right)^2(1-q_a)}{W_a} \right.$$
$$\left. + \sum_{a=1}^{K}\frac{q_a\left(1 - (1 - \frac{1}{\gamma} - \eta d)q_a\right)^2}{W_a} \right\}$$

$$\leq \max_{q \in \Delta_K}\left\{ \frac{2}{\gamma} + \eta d + \frac{1}{\gamma}\sum_{a \notin D}\frac{\left(q_a + \frac{1}{\gamma K}\right)^2(1-q_a) + q_a\left((1-q_a) + \frac{1}{\gamma}q_a + \eta d q_a\right)^2}{W_a} \right.$$
$$\left. + \frac{1}{\gamma}\sum_{a \in D}\frac{\left(q_a + \frac{1}{\gamma K} + \eta\right)^2(1-q_a) + q_a\left((1-q_a) + \frac{1}{\gamma}q_a + \eta d q_a\right)^2}{W_a} \right\}$$

$$\leq \max_{q \in \Delta_K}\left\{ \frac{2}{\gamma} + \eta d + \frac{1}{\gamma}\sum_{a \notin D}\frac{2\left(q_a^2 + \frac{1}{\gamma^2 K^2}\right)(1-q_a) + 3q_a\left((1-q_a)^2 + \frac{q_a^2}{\gamma^2} + \eta^2 d^2 q_a^2\right)}{W_a} \right.$$
$$\left. + \frac{1}{\gamma}\sum_{a \in D}\frac{3\left(q_a^2 + \frac{1}{\gamma^2 K^2} + \eta^2\right)(1-q_a) + 3q_a\left((1-q_a)^2 + \frac{q_a^2}{\gamma^2} + \eta^2 d^2 q_a^2\right)}{W_a} \right\}. \quad (17)$$

Now consider $a \in D$. If $a \in S$, then we know that $W_a \geq \eta$; Otherwise, we know that this node can be observed by at least one node in $D$, meaning that $W_a \geq \eta$. Combining the two cases above, we know that

$$\frac{1}{\gamma}\sum_{a \in D}\frac{3\left(q_a^2 + \frac{1}{\gamma^2 K^2} + \eta^2\right)(1-q_a) + 3q_a\left((1-q_a)^2 + \frac{1}{\gamma^2}q_a^2 + \eta^2 d^2 q_a^2\right)}{W_a}$$

$$\leq \frac{3}{\eta\gamma} \sum_{a\in D} \left[\left(q_a^2 + \frac{1}{\gamma^2 K^2} + \eta^2\right)(1-q_a) + q_a\left((1-q_a)^2 + \frac{1}{\gamma^2}q_a^2 + \eta^2 d^2 q_a^2\right)\right]$$

$$\leq \frac{3}{\eta\gamma} \sum_{a\in D} \left[q_a - q_a^2 + \frac{1}{\gamma^2}q_a^3 + \eta^2 d^2 q_a^3 + \frac{1}{\gamma^2 K^2} + \eta^2\right]$$

$$\leq \mathcal{O}\left(\frac{1}{\eta\gamma} + \frac{d\eta}{\gamma} + \frac{1}{\eta\gamma^3 K}\right) \qquad\qquad (\eta \leq \tfrac{1}{4d} \text{ and } \gamma \geq 16d)$$

$$\leq \mathcal{O}\left(\frac{1}{\eta\gamma}\right), \tag{18}$$

where the last inequality is because $\eta \leq \frac{1}{4d}$ and $\gamma \geq 16d$. Consider $a \notin D$. Let $S_0$ be the set of nodes which either have a self loop or can be observed by all the other node. Recall that $S$ represents the set of nodes with a self-loop. Then similar to the derivation of Eq. (13), we know that for $a \in S_0$,

$$\frac{1}{\gamma} \sum_{a\notin D, a\in S_0} \frac{2\left(q_a^2 + \frac{1}{\gamma^2 K^2}\right)(1-q_a) + 3q_a\left((1-q_a)^2 + \frac{q_a^2}{\gamma^2} + \eta^2 d^2 q_a^2\right)}{W_a}$$

$$\leq \frac{1}{\gamma} \sum_{a\notin D, a\in S_0} \frac{2q_a^2(1-q_a) + 3q_a\left((1-q_a)^2 + \frac{q_a^2}{\gamma^2} + \eta^2 d^2 q_a^2\right)}{W_a} + \mathcal{O}\left(\frac{1}{\gamma^2} + \frac{1}{\eta\gamma^3 K}\right)$$

$$(W_a \geq \tfrac{1}{\gamma K} \text{ if } a \in S \text{ and } W_a \geq \eta \text{ if } a \in [K]\backslash S$$

$$\leq \mathcal{O}\left(\frac{1}{\gamma} \sum_{a\in S_0, a\notin D} \frac{q_a(1-q_a)}{W_a} + \frac{1}{\gamma^3}\sum_{a\in S, a\notin D} q_a^2 + \frac{1}{\gamma^3}\sum_{a\in S_0, a\notin D\cup S}\frac{q_a^3}{\frac{K-1}{\gamma K}} + \frac{1}{\gamma^2} + \frac{1}{\eta\gamma^3 K}\right)$$

$$+ \mathcal{O}\left(\frac{1}{\gamma}\sum_{a\in S, a\notin D}\eta^2 d^2 q_a^2 + \frac{1}{\gamma}\sum_{a\in S_0, a\notin D\cup S}\frac{\eta^2 d^2 q_a^3}{\eta}\right)$$

$$(\text{for } a \in S_0, a\notin S, W_a \geq \max\{\tfrac{K-1}{\gamma K}, \eta\})$$

$$\leq \mathcal{O}\left(\frac{1}{\gamma}\sum_{a\in S_0, a\notin D}\frac{q_a(1-q_a)}{W_a} + \frac{1}{\eta\gamma}\right). \tag{19}$$

For $a \notin S_0$, we know that $W_a \geq \eta$. Therefore,

$$\frac{1}{\gamma} \sum_{a\notin D\cup S_0} \frac{2\left(q_a^2 + \frac{1}{\gamma^2 K^2}\right)(1-q_a) + 3q_a\left((1-q_a)^2 + \frac{q_a^2}{\gamma^2} + \eta^2 d^2 q_a^2\right)}{W_a}$$

$$\leq \frac{1}{\gamma\eta} \sum_{a\notin D\cup S_0}\left(2\left(q_a^2 + \frac{1}{\gamma^2 K^2}\right)(1-q_a) + 3q_a\left((1-q_a)^2 + \frac{q_a^2}{\gamma^2} + \frac{1}{16}q_a^2\right)\right)$$

$$\leq \frac{1}{\gamma\eta} \sum_{a\notin D\cup S_0}\left(2q_a(1-q_a) + \frac{1}{\gamma^2 K^2} + \frac{2q_a^3}{\gamma^2} + \frac{3}{16}q_a^3\right)$$

$$\leq \mathcal{O}\left(\frac{1}{\gamma\eta}\right). \tag{20}$$

Plugging Eq. (18), Eq. (19), and Eq. (20) to Eq. (17), we obtain that

$$\overline{\mathsf{dec}}_\gamma(p; \widehat{f}, x, G) \leq \mathcal{O}\left(\frac{1}{\gamma} + \eta d + \frac{1}{\gamma\eta} + \frac{1}{\gamma}\sum_{a\in S_0, a\notin D}\frac{q_a(1-q_a)}{W_a}\right) \tag{21}$$

Consider the last term. If $a \in S_0\backslash S$, similar to Eq. (14), we know that

$$\frac{q_a(1-q_a)}{W_a} \leq \frac{q_a(1-q_a)}{1-(1-\frac{1}{\gamma}-d\eta)q_a - \frac{1}{\gamma K}} \leq \frac{q_a(1-q_a)}{(1-\frac{1}{\gamma}-\eta d)(1-q_a)} \leq \frac{1}{1-\frac{1}{\gamma}-\eta d}q_a \leq \mathcal{O}(q_a),$$

where the last inequality is due to $\gamma \geq 16d$ and $\eta \leq \frac{1}{4d}$. If $a \in S$, similar to Eq. (15), we know that

$$\sum_{a \in S} \frac{q_a(1-q_a)}{W_a} \leq \sum_{a \in S} \frac{q_a(1-q_a)}{\sum_{j \in N^{\text{in}}(G,a)}((1-\frac{1}{\gamma}-\eta d)q_j + \frac{1}{\gamma K})}$$

$$\leq \frac{\gamma}{\gamma - 1 - \gamma \eta d} \sum_{a \in S} \frac{((1-\frac{1}{\gamma}-\eta d)q_a + \frac{1}{\gamma K})(1-q_a)}{\sum_{j \in N^{\text{in}}(G,a)}((1-\frac{1}{\gamma}-\eta d)q_j + \frac{1}{\gamma K})}$$

$$\leq 2 \sum_{a \in S} \frac{\left((1-\frac{1}{\gamma}-\eta d)q_a + \frac{1}{\gamma K}\right)}{\sum_{j \in N^{\text{in}}(G,a)}\left((1-\frac{1}{\gamma}-\eta d)q_j + \frac{1}{\gamma K}\right)} \qquad (\gamma \geq 4, \eta \leq \frac{1}{4d})$$

$$\leq \mathcal{O}(\widetilde{\alpha} \log(K\gamma)), \tag{22}$$

where the last inequality is again due to Lemma 5 in [Alon et al., 2015] and $\widetilde{\alpha}$ is the independence number of the subgraph induced by nodes with self-loops in $G$. Plugging Eq. (22) to Eq. (21) gives

$$\overline{\text{dec}}_\gamma(p; \widehat{f}, x, G) \leq \mathcal{O}\left(\eta d + \frac{1}{\gamma \eta} + \frac{\widetilde{\alpha} \log(K\gamma)}{\gamma}\right).$$

Picking $\eta = \sqrt{\frac{1}{\gamma d}} \leq \frac{1}{4d}$ proves the result. $\qquad \square$

Next, we prove Corollary 3.5 by combining Theorem 3.4 and Theorem 3.1.

**Corollary 3.5.** *Suppose that $G_t$ is deterministic, weakly observable, and has weak domination number no more than $d$ for all $t \in [T]$. In addition, suppose that the independence number of the subgraph induced by nodes with self-loops in $G_t$ is no more than $\widetilde{\alpha}$ for all $t \in [T]$. Then, Algorithm 1 with $\gamma = \max\{16d, \sqrt{\widetilde{\alpha}T/\mathbf{Reg}_{\text{Sq}}}, d^{\frac{1}{3}}T^{\frac{2}{3}}\mathbf{Reg}_{\text{Sq}}^{-\frac{2}{3}}\}$ guarantees that*

$$\mathbb{E}[\mathbf{Reg}_{\text{CB}}] \leq \widetilde{\mathcal{O}}\left(d^{\frac{1}{3}}T^{\frac{2}{3}}\mathbf{Reg}_{\text{Sq}}^{\frac{1}{3}} + \sqrt{\widetilde{\alpha}T\mathbf{Reg}_{\text{Sq}}}\right).$$

*Proof.* Combining Eq. (8), Eq. (9) and Theorem 3.4, we can bound $\mathbf{Reg}_{\text{CB}}$ as follows:

$$\mathbb{E}[\mathbf{Reg}_{\text{CB}}] \leq \mathcal{O}\left(\sqrt{\frac{d}{\gamma}}T + \frac{\widetilde{\alpha}T \log(K\gamma)}{\gamma} + \gamma\mathbf{Reg}_{\text{CB}}\right).$$

Picking $\gamma = \max\left\{16d, \sqrt{\widetilde{\alpha}T/\mathbf{Reg}_{\text{Sq}}}, d^{\frac{1}{3}}T^{\frac{2}{3}}\mathbf{Reg}_{\text{Sq}}^{-\frac{2}{3}}\right\}$ finishes the proof. $\qquad \square$

### A.3 Python Solution to Eq. (5)

```python
def makeProblem(nactions):
    import cvxpy as cp

    sqrtgammaG = cp.Parameter((nactions, nactions), nonneg=True)
    sqrtgammafhat = cp.Parameter(nactions)
    p = cp.Variable(nactions, nonneg=True)
    sqrtgammaz = cp.Variable()
    objective = cp.Minimize(sqrtgammafhat @ p + sqrtgammaz)
    constraints = [
        cp.sum(p) == 1
    ] + [
        cp.sum([ cp.quad_over_lin(eai - pi, vi)
                 for i, (pi, vi) in enumerate(zip(p, v))
                 for eai in (1 if i == a else 0,)
               ]) <= sqrtgammafhata + sqrtgammaz
        for v in (sqrtgammaG @ p,)
        for a, sqrtgammafhata in enumerate(sqrtgammafhat)
    ]
    problem = cp.Problem(objective, constraints)
    assert problem.is_dcp(dpp=True) # proof of convexity
    return problem, sqrtgammaG, sqrtgammafhat, p, sqrtgammaz
```

This particular formulation multiplies both sides of the constraint in Eq. (5) by $\sqrt{\gamma}$ while scaling the objective by $\sqrt{\gamma}$. While mathematically equivalent to Eq. (5), empirically it has superior numerical stability for large $\gamma$. For additional stability, when using this routine we recommend subtracting off the minimum value from $\widehat{f}$, which is equivalent to making the substitutions $\sqrt{\gamma}\widehat{f} \leftarrow \sqrt{\gamma}\widehat{f} - \sqrt{\gamma}\min_a \widehat{f}_a$ and $z \leftarrow z + \sqrt{\gamma}\min_a \widehat{f}_a$ and then exploiting the $\mathbf{1}^\top p = 1$ constraint.

### A.4   Proof of Theorem 3.6

**Theorem 3.6.** *Solving $\operatorname{argmin}_{p \in \Delta(K)} \overline{\mathsf{dec}}_\gamma(p; \widehat{f}, x, G)$ is equivalent to solving the following convex optimization problem.*

$$\min_{p \in \Delta(K), z} \quad p^\top \widehat{f} + z \tag{5}$$

$$\text{subject to} \quad \forall a \in [K] : \frac{1}{\gamma}\|p - e_a\|^2_{\operatorname{diag}(G^\top p)^{-1}} \leq \widehat{f}(x, a) + z,$$

$$G^\top p \succ 0,$$

*where $\widehat{f}$ in the objective is a shorthand for $\widehat{f}(x, \cdot) \in \mathbb{R}^K$, $e_a$ is the $a$-th standard basis vector, and $\succ$ means element-wise greater.*

*Proof.* Denote $f^\star = f^\star(x, \cdot) \in \mathbb{R}^K$. Note that according to the definition of $G$, we know that $(G^\top p)_i$ denotes the probability that action $i$'s loss is revealed when the selected action $a$ is sampled from distribution $p$. Then, we know that

$$\overline{\mathsf{dec}}_\gamma(p; \widehat{f}, x, G)$$

$$= \sup_{\substack{a^\star \in [K] \\ f^\star \in \mathbb{R}^K}} \mathbb{E}_{a_t \sim p}\left[ f^\star_{a_t} - f^\star_{a^\star} - \frac{\gamma}{4}\mathbb{E}_{A \sim G(\cdot | a_t)}\left[ \sum_{a \in A}\left(\widehat{f}_a - f^\star_a\right)^2 \right] \right]$$

$$= \sup_{\substack{a^\star \in [K] \\ f^\star \in \mathbb{R}^K}} (p - e_{a^\star})^\top f^\star - \frac{\gamma}{4} \sum_{a \in [K]} \|\widehat{f} - f^\star\|^2_{\operatorname{diag}(G^\top p)}$$

$$= \sup_{a^\star \in [K]} (p - e_{a^\star})^\top \widehat{f} + \frac{1}{\gamma}\|p - e_{a^\star}\|^2_{\operatorname{diag}(G^\top p)^{-1}} \qquad (G^\top p \succ 0)$$

$$= p^\top \widehat{f} + \max_{a^\star \in [K]}\left\{ \frac{1}{\gamma}\|p - e_{a^\star}\|^2_{\operatorname{diag}(G^\top p)^{-1}} - e_{a^\star}^\top \widehat{f} \right\},$$

where the third equality is by picking $f^\star$ to be the maximizer and introduces a constraint. Therefore, the minimization problem $\min_{p \in \Delta(K)} \overline{\mathsf{dec}}_\gamma(p; \widehat{f}, x, G)$ can be written as the following constrained optimization by variable substitution:

$$\min_{p \in \Delta(K), z} \quad p^\top \widehat{f} + z$$

$$\text{subject to} \quad \forall a \in [K] : \frac{1}{\gamma}\|p - e_a\|^2_{\operatorname{diag}(G^\top p)^{-1}} \leq e_a^\top \widehat{f} + z,$$

$$G^\top p \succ 0.$$

The convexity of the constraints follows from Lemma A.1. $\qquad \square$

## B   Omitted Details in Section 4

In this section, we provide proofs for Section 4. We define $W_a := \sum_{a' \in N^{\mathrm{in}}(G, a)} p_{a'}$ to be the probability that the loss of action $a$ is revealed when selecting an action from distribution $p$. Let $\widehat{f} = \widehat{f}(x, \cdot) \in \mathbb{R}^K$ and $f = f(x, \cdot) \in \mathbb{R}^K$. Direct calculation shows that for any $a^\star \in [K]$,

$$f^\star = \operatorname*{argmax}_{f \in \mathbb{R}^K} \mathbb{E}_{a \sim p}\left[ f(x, a) - f(x, a^\star) - \frac{\gamma}{4} \cdot \sum_{a' \in N^{\mathrm{in}}(G, a)} (\widehat{f}_t(x, a') - f(x, a'))^2 \right]$$

$$= \frac{2}{\gamma} \operatorname{diag}(W)^{-1}(p - e_{a^\star}) + \widehat{f}.$$

Therefore, substituting $f^\star$ into Eq. (4), we obtain that

$$\overline{\operatorname{dec}}_\gamma(p; \widehat{f}, x, G) = \max_{a^\star \in [K]} \left\{ \frac{1}{\gamma} \left( \sum_{a=1}^K \frac{p_a^2}{W_a} + \frac{1 - 2p_{a^\star}}{W_{a^\star}} \right) + \left\langle p - e_{a^\star}, \widehat{f} \right\rangle \right\}. \tag{23}$$

Without loss of generality, we assume the $\min_{i \in [K]} \widehat{f}_i = 0$. This is because shifting $\widehat{f}$ by $\min_{i \in [K]} \widehat{f}_i$ does not change the value of $\left\langle p - e_{a^\star}, \widehat{f} \right\rangle$. In the following sections, we provide proofs showing that a certain closed-form of $p$ leads to optimal $\overline{\operatorname{dec}}_\gamma(p; \widehat{f}, x, G)$ up to constant factors for several specific types of feedback graphs, respectively.

### B.1 Cops-and-Robbers Graph

**Proposition 1.** *When $G = G_{\mathrm{CR}}$, given any $\widehat{f}$, context $x$, the closed-form distribution $p$ in Eq. (6) guarantees that $\overline{\operatorname{dec}}_\gamma(p; \widehat{f}, x, G_{\mathrm{CR}}) \leq \mathcal{O}\left(\frac{1}{\gamma}\right)$.*

*Proof.* We use the following notation for convenience: $p_1 := p_{a_1}$, $p_2 := p_{a_2}$, $\widehat{f}_1 := \widehat{f}_{a_1} = 0$, $\widehat{f}_2 := \widehat{f}_{a_2}$. For the cops-and-robbers graph and closed-form solution $p$ in Eq. (6), Eq. (23) becomes:

$$\overline{\operatorname{dec}}_\gamma(p; \widehat{f}, x, G_{\mathrm{CR}}) = \max_{a^\star \in [K]} \left\{ \frac{1}{\gamma} \left( \frac{p_1^2}{1 - p_1} + \frac{(1 - p_1)^2}{p_1} + \frac{1 - 2p_{a^\star}}{W_{a^\star}} \right) + \left\langle p - e_{a^\star}, \widehat{f} \right\rangle \right\}.$$

If $a^\star \neq a_1$ and $a^\star \neq a_2$, we know that

$$
\begin{aligned}
&\frac{1}{\gamma} \left( \frac{p_1^2}{1 - p_1} + \frac{(1 - p_1)^2}{p_1} + \frac{1 - 2p_{a^\star}}{W_{a^\star}} \right) + \left\langle p - e_{a^\star}, \widehat{f} \right\rangle \\
&= \frac{1}{\gamma} \left( \frac{p_1^2}{1 - p_1} + \frac{(1 - p_1)^2}{p_1} + 1 \right) + p_1 \widehat{f}_1 + p_2 \widehat{f}_2 - \widehat{f}_{a^\star} \\
&\leq \frac{1}{\gamma} \left( \frac{p_1^2}{1 - p_1} + \frac{(1 - p_1)^2}{p_1} + 1 \right) - p_1 \widehat{f}_2 && (\widehat{f}_{a^\star} \geq \widehat{f}_2 \geq \widehat{f}_1 = 0) \\
&\leq \frac{1}{\gamma} \left( \frac{1}{1 - p_1} + 1 + 1 \right) - p_1 \widehat{f}_2 && (p_1 \in [\tfrac{1}{2}, 1], p_1 \geq p_2 \in [0, \tfrac{1}{2}]) \\
&= \frac{1}{\gamma} \left( 4 + \gamma \widehat{f}_2 \right) - \left( 1 - \frac{1}{2 + \gamma \widehat{f}_2} \right) \widehat{f}_2 \\
&\leq \frac{5}{\gamma}.
\end{aligned}
$$

If $a^\star = a_2$, we can obtain that

$$
\begin{aligned}
&\frac{1}{\gamma} \left( \frac{p_1^2}{1 - p_1} + \frac{(1 - p_1)^2}{p_1} + \frac{1 - 2p_{a^\star}}{W_{a^\star}} \right) + \left\langle p - e_{a^\star}, \widehat{f} \right\rangle \\
&= \frac{1}{\gamma} \left( \frac{p_1^2}{1 - p_1} + \frac{(1 - p_1)^2}{p_1} + \frac{1 - 2p_2}{p_1} \right) + p_1 \widehat{f}_1 + p_2 \widehat{f}_2 - \widehat{f}_2 \\
&\leq \frac{1}{\gamma} \left( \frac{p_1^2}{1 - p_1} + \frac{(1 - p_1)^2}{p_1} + \frac{1 - 2(1 - p_1)}{p_1} \right) - p_1 \widehat{f}_2 && (\widehat{f}_1 = 0) \\
&\leq \frac{1}{\gamma} \left( \frac{1}{1 - p_1} + 1 + 2 - \frac{1}{p_1} \right) - p_1 \widehat{f}_2 && (p_1 \in [\tfrac{1}{2}, 1], p_2 \in [0, \tfrac{1}{2}]) \\
&\leq \frac{1}{\gamma} \left( 5 + \gamma \widehat{f}_2 \right) - \left( 1 - \frac{1}{2 + \gamma \widehat{f}_2} \right) \widehat{f}_2 && (p_1 = \tfrac{1}{2 + \gamma \widehat{f}_2}) \\
&\leq \frac{6}{\gamma}.
\end{aligned}
$$

If $a^\star = a_1$, we have

$$\frac{1}{\gamma}\left(\frac{p_1^2}{1-p_1} + \frac{(1-p_1)^2}{p_1} + \frac{1-2p_{a^\star}}{W_{a^\star}}\right) + \left\langle p - e_{a^\star}, \widehat{f}\right\rangle$$

$$\leq \frac{1}{\gamma}\left(\frac{p_1^2}{1-p_1} + \frac{(1-p_1)^2}{p_1} + \frac{1-2p_1}{1-p_1}\right) + (1-p_1)\widehat{f}_2$$

$$\leq \frac{1}{\gamma}\left(1-p_1 + \frac{(1-p_1)^2}{p_1}\right) + (1-p_1)\widehat{f}_2$$

$$\leq \frac{1}{\gamma}\left(1 + \frac{1}{2}\right) + \frac{\widehat{f}_2}{2+\gamma\widehat{f}_2} \qquad\qquad (p_1 \in [\tfrac{1}{2}, 1])$$

$$\leq \frac{3}{\gamma}.$$

Putting everything together, we prove that $\overline{\mathsf{dec}}_\gamma(p; \widehat{f}, x, G_{\mathrm{CR}}) \leq \frac{6}{\gamma} \leq \mathcal{O}\left(\frac{1}{\gamma}\right)$. $\qquad\square$

## B.2 Apple Tasting Graph

**Proposition 2.** *When $G = G_{\mathrm{AT}}$, given any $\widehat{f}$, context $x$, the closed-form distribution $p$ in Eq. (7) guarantees that $\overline{\mathsf{dec}}_\gamma(p; \widehat{f}, x, G_{\mathrm{AT}}) \leq \mathcal{O}(\frac{1}{\gamma})$.*

*Proof.* For the apple tasting graph and closed-form solution $p$ in Eq. (7), Eq. (23) becomes:

$$\overline{\mathsf{dec}}_\gamma(p; \widehat{f}, x, G) = \max_{a^\star \in [K]}\left\{\frac{1}{\gamma}\left(p_1 + \frac{(1-p_1)^2}{p_1} + \frac{1-2p_{a^\star}}{W_{a^\star}}\right) + \left\langle p - e_{a^\star}, \widehat{f}\right\rangle\right\}.$$

Suppose $\widehat{f}_1 = 0$, we know that $p_1 = 1$, $p_2 = 0$ and

1. If $a^\star = 1$, we have

$$\frac{1}{\gamma}\left(p_1 + \frac{(1-p_1)^2}{p_1} + \frac{1-2p_{a^\star}}{W_{a^\star}}\right) + \left\langle p - e_{a^\star}, \widehat{f}\right\rangle = 0.$$

2. If $a^\star = 2$, direct calculation shows that

$$\frac{1}{\gamma}\left(p_1 + \frac{(1-p_1)^2}{p_1} + \frac{1-2p_{a^\star}}{W_{a^\star}}\right) + \left\langle p - e_{a^\star}, \widehat{f}\right\rangle \leq \frac{2}{\gamma}.$$

Suppose $\widehat{f}_2 = 0$, we know that $p_1 = \frac{2}{4+\gamma\widehat{f}_1}$, $p_2 = 1 - p_1$ and

1. If $a^\star = 1$, we have

$$\frac{1}{\gamma}\left(p_1 + \frac{(1-p_1)^2}{p_1} + \frac{1-2p_{a^\star}}{W_{a^\star}}\right) + \left\langle p - e_{a^\star}, \widehat{f}\right\rangle$$

$$= \frac{1}{\gamma}\left(p_1 + \frac{(1-p_1)^2}{p_1} + \frac{1-2p_1}{p_1}\right) - (1-p_1)\widehat{f}_1$$

$$= \frac{2(1-p_1)^2}{\gamma p_1} - (1-p_1)\widehat{f}_1$$

$$= \frac{(2+\gamma\widehat{f}_1)^2}{\gamma(4+\gamma\widehat{f}_1)} - (1-p_1)\widehat{f}_1$$

$$\leq \frac{4+\gamma\widehat{f}_1}{\gamma} + \frac{2\widehat{f}_1}{4+\gamma\widehat{f}_1} - \widehat{f}_1 \leq \frac{6}{\gamma}.$$

2. If $a^\star = 2$, direct calculation shows that

$$\frac{1}{\gamma}\left(p_1 + \frac{(1-p_1)^2}{p_1} + \frac{1-2p_{a^\star}}{W_{a^\star}}\right) + \left\langle p - e_{a^\star}, \widehat{f}\right\rangle = \frac{2p_1}{\gamma} + p_1\widehat{f}_1 \leq \frac{1}{\gamma} + \frac{2\widehat{f}_1}{4+\gamma\widehat{f}_1} \leq \frac{3}{\gamma}.$$

Putting everything together, we prove that $\overline{\mathsf{dec}}_\gamma(p; \widehat{f}, x, G_{\mathrm{AT}}) \leq \frac{6}{\gamma} \leq \mathcal{O}\left(\frac{1}{\gamma}\right)$. $\qquad\qquad$ □

## B.3 Inventory Graph

**Proposition 3.** *When $G = G_{\mathrm{inv}}$, given any $\widehat{f}$, context $x$, there exists a closed-form distribution $p \in \Delta(K)$ guaranteeing that $\overline{\mathsf{dec}}_\gamma(p; \widehat{f}, x, G_{\mathrm{inv}}) \leq \mathcal{O}(\frac{1}{\gamma})$, where $p$ is defined as follows: $p_j = \max\{\frac{1}{1+\gamma(\widehat{f}_j - \min_i \widehat{f}_i)} - \sum_{j' > j} p_{j'}, 0\}$ for all $j \in [K]$.*

*Proof.* Based on the distribution defined above, define $A \subseteq [K]$ to be the set such that for all $i \in A$, $p_i > 0$ and denote $N = |A|$. We index each action in $A$ by $k_1 < k_2 < \cdots < k_N = K$. According to the definition of $p_i$, we know that $p_i$ is strictly positive only when $\widehat{f}_i < \widehat{f}_j$ for all $j > i$ and specifically, when $p_i > 0$, we know that $W_i = \sum_{j \geq i} p_j = \frac{1}{1+\gamma \widehat{f}_i}$ (recall that $\min_i \widehat{f}_i = 0$ since we shift $\widehat{f}$). Therefore, define $W_{k_{N+1}} = 0$ and we know that

$$
\begin{aligned}
&\overline{\mathsf{dec}}_\gamma(p; \widehat{f}, x, G_{\mathrm{inv}}) \\
&= \sum_{i=1}^{N} p_{k_i} \widehat{f}_{k_i} + \frac{1}{\gamma} \sum_{a=1}^{K} \frac{p_a^2}{W_a} + \max_{a^\star \in [K]} \left\{ \frac{1 - 2p_{a^\star}}{\gamma W_{a^\star}} - \widehat{f}_{a^\star} \right\} \\
&\leq \sum_{i=1}^{N} \left( W_{k_i} - W_{k_{i+1}} \right) \widehat{f}_{k_i} + \frac{1}{\gamma} + \max_{a^\star \in [K]} \left\{ \frac{1 - 2p_{a^\star}}{\gamma W_{a^\star}} - \widehat{f}_{a^\star} \right\} \\
&\leq \frac{2}{\gamma} + \sum_{i=1}^{N-1} \left( \frac{1}{1+\gamma \widehat{f}_{k_i}} - \frac{1}{1+\gamma \widehat{f}_{k_{i+1}}} \right) \widehat{f}_{k_i} + \max_{a^\star \in [K]} \left\{ \frac{1 - 2p_{a^\star}}{\gamma W_{a^\star}} - \widehat{f}_{a^\star} \right\} \\
&\leq \frac{3}{\gamma} + \sum_{i=2}^{N} \frac{\widehat{f}_{k_i} - \widehat{f}_{k_{i-1}}}{1+\gamma \widehat{f}_{k_i}} + \max_{a^\star \in [K]} \left\{ \frac{1 - 2p_{a^\star}}{\gamma W_{a^\star}} - \widehat{f}_{a^\star} \right\}.
\end{aligned}
$$

According to Lemma 9 of [Alon et al., 2013] (included as Lemma E.2 for completeness), we know that

$$
\sum_{i=2}^{N} \frac{\widehat{f}_{k_i} - \widehat{f}_{k_{i-1}}}{1+\gamma \widehat{f}_{k_i}} = \frac{1}{\gamma} \sum_{i=2}^{N} \frac{\widehat{f}_{k_i} - \widehat{f}_{k_{i-1}}}{\frac{1}{\gamma} + \widehat{f}_{k_i}} \leq \frac{\mathrm{mas}(G_A)}{\gamma} = \frac{1}{\gamma}, \tag{24}
$$

where $G_A$ is the subgraph of $G$ restricted to node set $A$ and $\mathrm{mas}(G)$ is the size of the maximum acyclic subgraphs of $G$. It is direct to see that any subgraph $G$ of $G_{\mathrm{inv}}$ has $\mathrm{mas}(G) = 1$.

Next, consider the value of $a^\star \in [K]$ that maximizes $\frac{1 - 2p_{a^\star}}{\gamma W_{a^\star}} - \widehat{f}_{a^\star}$. If $a^\star \leq k_1$, then we know that $W_{a^\star} = 1$ and $\frac{1 - 2p_{a^\star}}{\gamma W_{a^\star}} - \widehat{f}_{a^\star} \leq \frac{1}{\gamma}$. Otherwise, suppose that $k_i < a^\star \leq k_{i+1}$ for some $i \in [N-1]$. According to the definition of $p$, if $a^\star \neq k_{i+1}$ we know that $p_{a^\star} = 0$ and

$$
\frac{1}{1+\gamma \widehat{f}_{a^\star}} \leq \sum_{j' > a^\star} p_{j'} = W_{k_{i+1}} = W_{a^\star}.
$$

Therefore,

$$
\frac{1 - 2p_{a^\star}}{\gamma W_{a^\star}} - \widehat{f}_{a^\star} = \frac{1}{\gamma W_{a^\star}} - \widehat{f}_{a^\star} \leq \frac{1}{\gamma}.
$$

Otherwise, $W_{a^\star} = W_{k_{i+1}}$ and $\frac{1 - 2p_{a^\star}}{\gamma W_{a^\star}} - \widehat{f}_{a^\star} \leq \frac{1}{\gamma W_{k_{i+1}}} - \widehat{f}_{k_{i+1}} = \frac{1}{\gamma}$. Combining the two cases above and Eq. (24), we obtain that

$$
\overline{\mathsf{dec}}_\gamma(p; \widehat{f}, x, G_{\mathrm{inv}}) \leq \frac{3}{\gamma} + \frac{1}{\gamma} + \frac{1}{\gamma} = \mathcal{O}\left(\frac{1}{\gamma}\right).
$$

$\qquad\qquad$ □

## B.4 Undirected and Self-Aware Graphs

**Proposition 4.** *When $G$ is an undirected self-aware graph, given any $\widehat{f}$, context $x$, there exists a closed-form distribution $p \in \Delta(K)$ guaranteeing that $\overline{\mathsf{dec}}_\gamma(p; \widehat{f}, x, G) \leq \mathcal{O}\left(\frac{\alpha}{\gamma}\right)$.*

*Proof.* We first introduce the closed-form of $p$ and then show that $\overline{\mathsf{dec}}_\gamma(p; \widehat{f}, x, G) \leq \mathcal{O}(\frac{\alpha}{\gamma})$. Specifically, we first sort $\widehat{f}_a$ in an increasing order and choose a maximal independent set by choosing the nodes in a greedy way. Specifically, we pick $k_1 = \operatorname{argmin}_{i \in [K]} \widehat{f}_i$. Then, we ignore all the nodes that are connected to $k_1$ and select the node $a$ with the smallest $\widehat{f}_a$ in the remaining node set. This forms a maximal independent set $I \subseteq [K]$, which has size no more than $\alpha$ and is also a dominating set. Set $p_a = \frac{1}{\alpha + \gamma \widehat{f}_a}$ for $a \in I \backslash \{k_1\}$ and $p_{k_1} = 1 - \sum_{a \neq k_1, a \in I} p_a$. This is a valid distribution as we only choose at most $\alpha$ nodes and $p_a \leq 1/\alpha$ for all $a \in I \backslash \{k_1\}$. Now we show that $\overline{\mathsf{dec}}_\gamma(p; \widehat{f}, x, G) \leq \mathcal{O}(\frac{\alpha}{\gamma})$. Specifically, we only need to show that with this choice of $p$, for any $a^\star \in [K]$,

$$\sum_{a=1}^K p_a \widehat{f}_a - \widehat{f}_{a^\star} + \frac{1}{\gamma} \sum_{a=1}^K \frac{p_a^2}{W_a} + \frac{1 - 2p_{a^\star}}{\gamma W_{a^\star}} \leq \mathcal{O}\left(\frac{\alpha}{\gamma}\right).$$

Plugging in the form of $p$, we know that

$$\sum_{a=1}^K p_a \widehat{f}_a - \widehat{f}_{a^\star} + \frac{1}{\gamma} \sum_{a=1}^K \frac{p_a^2}{W_a} + \frac{1 - 2p_{a^\star}}{\gamma W_{a^\star}}$$
$$\leq \sum_{a \in I \backslash \{k_1\}} \frac{\widehat{f}_a}{\alpha + \gamma \widehat{f}_a} - \widehat{f}_{a^\star} + \frac{1 - 2p_{a^\star}}{\gamma W_{a^\star}} + \frac{1}{\gamma} \qquad (p_a \leq W_a \text{ for all } a \in [K])$$
$$\leq \frac{\alpha}{\gamma} - \widehat{f}_{a^\star} + \frac{1 - 2p_{a^\star}}{\gamma W_{a^\star}}. \qquad (|I| \leq \alpha)$$

If $a^\star = k_1$, then we can obtain that $\frac{1 - 2p_{a^\star}}{\gamma W_{a^\star}} \leq \frac{1}{\gamma W_{k_1}} \leq \frac{\alpha}{\gamma}$ as $p_{k_1} \geq \frac{1}{\alpha}$ according to the definition of $p$. Otherwise, note that according to the choice of the maximal independent set $I$, $W_{a^\star} \geq \frac{1}{\alpha + \gamma \widehat{f}_{a'}}$ for some $a' \in I$ such that $\widehat{f}_{a'} \leq \widehat{f}_{a^\star}$. Therefore,

$$-\widehat{f}_{a^\star} + \frac{1 - 2p_{a^\star}}{\gamma W_{a^\star}} \leq -\widehat{f}_{a^\star} + \frac{1}{\gamma W_{a^\star}} \leq -\widehat{f}_{a^\star} + \frac{\alpha + \gamma \widehat{f}_{a'}}{\gamma} \leq \frac{\alpha}{\gamma}.$$

Combining the two inequalities above together proves the bound. $\qquad \square$

# C  Implementation Details in Experiments

## C.1  Implementation Details in Section 5.1

We conduct experiments on RCV1 [Lewis et al., 2004], which is a multilabel text-categorization dataset. We use a subset of RCV1 containing 50000 samples and $K = 50$ sub-classes. Therefore, the feedback graph in our experiment has $K = 50$ nodes. We use the bag-of-words vector of each sample as the context with dimension $d = 47236$ and treat the text categories as the arms. In each round $t$, the learner receives the bag-of-words vector $x_t$ and makes a prediction $a_t \in [K]$ as the text category. The loss is set to be $\ell_{t,a_t} = 0$ if the sample belongs to the predicted category $a_t$ and $\ell_{t,a_t} = 1$ otherwise.

The function class we consider is the following linear function class:

$$\mathcal{F} = \{f : f(x, a) = \mathsf{Sigmoid}((Mx)_a), M \in \mathbb{R}^{K \times d}\},$$

where $\mathsf{Sigmoid}(u) = \frac{1}{1 + e^{-u}}$ for any $u \in \mathbb{R}$. The oracle is implemented by applying online gradient descent with learning rate $\eta$ searched over $\{0.1, 0.2, 0.5, 1, 2, 4\}$. As suggested by [Foster and

Krishnamurthy, 2021], we use a time-varying exploration parameter $\gamma_t = c \cdot \sqrt{\alpha t}$, where $t$ is the index of the iteration, $c$ is searched over $\{8, 16, 32, 64, 128\}$, and $\alpha$ is the independence number of the corresponding feedback graph. Our code is built on PyTorch framework [Paszke et al., 2019]. We run 5 independent experiments with different random seeds and plot the mean and standard deviation value of PV loss.

## C.2    Implementation Details in Section 5.2

### C.2.1    Details for Results on Random Directed Self-aware Graphs

We conduct experiments on a subset of RCV1 containing 10000 samples with $K = 10$ sub-classes. Our code is built on Vowpal Wabbit [Langford and Zhang, 2007]. For SqaureCB, the exploration parameter $\gamma_t$ at round $t$ is set to be $\gamma_t = c \cdot \sqrt{Kt}$, where $t$ is the index of the round and $c$ is the hyper-parameter searched over set $\{8, 16, 32, 64, 128\}$. The remaining details are the same as described in Appendix C.1.

### C.2.2    Details for Results on Synthetic Inventory Dataset

In this subsection, we introduce more details in the synthetic inventory data construction, loss function constructions, oracle implementation, and computation of the strategy at each round.

**Dataset.**    In this experiment, we create a synthetic inventory dataset constructed as follows. The dataset includes $T$ data points, the $t$-th of which is represented as $(x_t, d_t)$ where $x_t \in \mathbb{R}^m$ is the context and $d_t$ is the realized demand given context $x_t$. Specifically, in the experiment, we choose $m = 100$ and $x_t$'s are drawn i.i.d from Gaussian distribution with mean 0 and standard deviation 0.1. The demand $d_t$ is defined as

$$d_t = \frac{1}{\sqrt{m}} x_t^\top \theta + \varepsilon_t,$$

where $\theta \in \mathbb{R}^m$ is an arbitrary vector and $\varepsilon_t$ is a one-dimensional Gaussian random variable with mean 0.3 and standard deviation 0.1. After all the data points $\{(x_t, d_t)\}_{t=1}^T$ are constructed, we normalize $d_t$ to $[0, 1]$ by setting $d_t \leftarrow \frac{d_t - \min_{t' \in [T]} d_{t'}}{\max_{t' \in [T]} d_{t'} - \min_{t' \in [T]} d_{t'}}$. In all our experiments, we set $T = 10000$.

**Loss construction.**    Next, we define the loss at round $t$ when picking the inventory level $a_t$ with demand $d_t$, which is defined as follows:

$$\ell_{t, a_t} = h \cdot \max\{a_t - d_t, 0\} + b \cdot \max\{d_t - a_t, 0\}, \tag{25}$$

where $h > 0$ is the holding cost per remaining items and $b > 0$ is the backorder cost per remaining items. In the experiment, we set $h = 0.25$ and $b = 1$.

**Regression oracle.**    The function class we use in this experiment is as follows:

$$\mathcal{F} = \{f : f(x, a) = h \cdot \max\{a - (x^\top \theta + \beta), 0\} + b \cdot \max\{x^\top \theta + \beta - a, 0\}, \theta \in \mathbb{R}^m, \beta \in \mathbb{R}\}.$$

This ensures the realizability assumption according to the definition of our loss function shown in Eq. (25). The oracle uses online gradient descent with learning rate $\eta$ searched over $\{0.01, 0.05, 0.1, 0.5, 1\}$.

**Calculation of $p_t$.**    To make SquareCB.G more efficient, instead of solving the convex program defined in Eq. (5), we use the closed-form of $p_t$ derived in Proposition 3, which only requires $\mathcal{O}(K)$ computational cost and has the same theoretical guarantee (up to a constant factor) as the one enjoyed by the solution solved by Eq. (5). Similar to the case in Appendix C.1, at each round $t$, we pick $\gamma_t = c \cdot \sqrt{t}$ with $c$ searched over the set $\{0.25, 0.5, 1, 2, 3, 4\}$. Note again that the independence number for inventory graph is 1.

We run 8 independent experiments with different random seeds and plot the mean and standard deviation value of PV loss.

# D Adaptive Tuning of $\gamma$ without the Knowledge of Graph-Theoretic Numbers

In this section, we show how to adaptively tune the parameter $\gamma$ in order to achieve $\widetilde{\mathcal{O}}\left(\sqrt{\sum_{t=1}^{T} \alpha_t \mathbf{Reg_{Sq}}}\right)$ regret in the strongly observable graphs case and $\widetilde{\mathcal{O}}\left(\left(\sum_{t=1}^{T} \sqrt{d_t}\right)^{\frac{2}{3}} \mathbf{Reg_{Sq}^{\frac{1}{3}}} + \sqrt{\sum_{t=1}^{T} \widetilde{\alpha}_t \mathbf{Reg_{Sq}}}\right)$ in the weakly observable graphs case.

## D.1 Strongly Observable Graphs

In order to achieve $\widetilde{\mathcal{O}}\left(\sum_{t=1}^{T} \alpha_t \mathbf{Reg_{Sq}}\right)$ regret guarantee without the knowledge of $\alpha_t$, we apply a doubling trick on $\gamma$ based on the value of $\min_{p \in \Delta(K)} \overline{\mathsf{dec}}_\gamma(p; \widehat{f}_t, x_t, G_t)$. Specifically, our algorithm goes in epochs with the parameter $\gamma$ being $\gamma_s$ in the $s$-th epoch. We initialize $\gamma_1 = \sqrt{\frac{T}{\mathbf{Reg_{Sq}}}}$. As proven in Theorem 3.2, we know that

$$\gamma \cdot \min_{p \in \Delta(K)} \overline{\mathsf{dec}}_\gamma(p; \widehat{f}_t, x_t, G_t) \le \widetilde{\mathcal{O}}(\alpha_t).$$

Therefore, within each epoch $s$ (with start round $b_s$), at round $t$, we calculate the value

$$Q_t = \sum_{\tau=b_s}^{t} \min_{p \in \Delta(K)} \overline{\mathsf{dec}}_{\gamma_s}(p; \widehat{f}_t, x_t, G_t), \tag{26}$$

which is bounded by $\widetilde{\mathcal{O}}\left(\frac{1}{\gamma_s} \sum_{\tau=b_s}^{t} \alpha_\tau\right)$ and is in fact obtainable by solving the convex program. Then, we check whether $Q_t \le \gamma_s \mathbf{Reg_{Sq}}$. If this holds, we continue our algorithm using $\gamma_s$; otherwise, we set $\gamma_{s+1} = 2\gamma_s$ and restart the algorithm.

Now we analyze the performance of the above described algorithm. First, note that for any $t$, within epoch $s$,

$$\gamma_s Q_t \le \widetilde{\mathcal{O}}\left(\sum_{\tau=1}^{T} \alpha_\tau\right),$$

meaning that the number of epoch $S$ is bounded by $S = \log_2 C_1 + \log_4 \frac{\sum_{t=1}^{T} \alpha_t}{T}$ for certain $C_1 > 0$ which only contains constant and log terms.

Next, consider the regret in epoch $s$ with $I_s = [b_s, e_s]$. According to Eq. (8), we know that the regret within epoch $s$ is bounded as follows:

$$\mathbb{E}\left[\sum_{t \in I_s} f^\star(x_t, a_t) - \sum_{t \in I_s} f^\star(x_t, \pi^\star(x_t))\right]$$

$$\le \mathbb{E}\left[\sum_{t \in I_s} \overline{\mathsf{dec}}_{\gamma_s}(p_t; \widehat{f}_t, x_t, G_t)\right] + \frac{\gamma_s}{4} \mathbf{Reg_{Sq}}$$

$$\le \mathbb{E}\left[\sum_{t \in [b_s, e_s - 1]} \overline{\mathsf{dec}}_{\gamma_s}(p_t; \widehat{f}_t, x_t, G_t)\right] + \widetilde{\mathcal{O}}\left(\frac{\alpha_{e_s}}{\gamma_s}\right) + \frac{\gamma_s}{4} \mathbf{Reg_{Sq}}$$

$$\le \widetilde{\mathcal{O}}(\gamma_s \mathbf{Reg_{Sq}}), \tag{27}$$

where the last inequality is because at round $t = e_s - 1$, $Q_t \le \gamma_s \mathbf{Reg_{Sq}}$ is satisfied. Taking summation over all $S$ epochs, we know that the overall regret is bounded as

$$\mathbb{E}[\mathbf{Reg_{CB}}] \le \sum_{s=1}^{S} \widetilde{\mathcal{O}}(\gamma_s \mathbf{Reg_{Sq}}) = \sum_{s=1}^{S} \widetilde{\mathcal{O}}\left(2^{s-1} \sqrt{T \mathbf{Reg_{Sq}}}\right)$$

$$\le \widetilde{\mathcal{O}}\left(2^S \sqrt{T \mathbf{Reg_{Sq}}}\right) = \widetilde{\mathcal{O}}\left(\sqrt{\sum_{t=1}^{T} \alpha_t \mathbf{Reg_{Sq}}}\right), \tag{28}$$

which finishes the proof.

### D.2 Weakly Observable Graphs

For the weakly observable graphs case, to achieve the target regret without the knowledge of $\widetilde{\alpha}_t$ and $d_t$, which are the independence number of the subgraph induced by nodes with self-loops in $G_t$ and the weak domination number of $G_t$, we apply the same approach as the one applied in the strongly observable graph case. Note that according to Theorem 3.4, within epoch $s$, we have $Q_t \leq C_2 \left( \frac{\sum_{\tau=b_s}^t \sqrt{d_\tau}}{\sqrt{\gamma_s}} + \frac{\sum_{\tau=b_s}^t \widetilde{\alpha}_\tau}{\gamma_s} \right)$ for certain $C_2 > 1$ only containing constants and $\log$ factors. In the weakly observable graphs case, we know that the number of epoch is bounded by

$$S = 2 + \log_2 C_2 + \max \left\{ \log_4 \frac{\sum_{t=1}^T \widetilde{\alpha}_t}{T}, \log_2 \left( \frac{(\sum_{t=1}^T \sqrt{d_t})^{\frac{2}{3}}}{\sqrt{T} \cdot \mathbf{Reg}_{\mathsf{Sq}}^{\frac{1}{6}}} \right) \right\} \text{ since we have}$$

$$\gamma_S = 4C_2 \cdot \sqrt{\frac{1}{\mathbf{Reg}_{\mathsf{Sq}}}} \max \left\{ \sqrt{\sum_{t=1}^T \widetilde{\alpha}_t}, \frac{(\sum_{t=1}^T \sqrt{d_t})^{\frac{2}{3}}}{\mathbf{Reg}_{\mathsf{Sq}}^{\frac{1}{6}}} \right\},$$

and at round $t$ in epoch $S$,

$$Q_t \leq C_2 \left( \frac{\sum_{\tau=1}^T \widetilde{\alpha}_\tau}{\gamma_S} + \frac{\sum_{\tau=1}^T \sqrt{d_\tau}}{\sqrt{\gamma_S}} \right) \leq \gamma_S \mathbf{Reg}_{\mathsf{Sq}},$$

meaning that epoch $S$ will never end. Therefore, following Eq. (27) and Eq. (28), we can obtain that

$$\mathbb{E}[\mathbf{Reg}_{\mathsf{CB}}] \leq \widetilde{\mathcal{O}}\left( 2^S \sqrt{T \mathbf{Reg}_{\mathsf{Sq}}} \right) = \widetilde{\mathcal{O}}\left( \sqrt{\sum_{t=1}^T \widetilde{\alpha}_t \mathbf{Reg}_{\mathsf{Sq}}} + \left( \sum_{t=1}^T \sqrt{d_t} \right)^{\frac{2}{3}} \mathbf{Reg}_{\mathsf{Sq}}^{\frac{1}{3}} \right),$$

which finishes the proof.

## E  Auxiliary Lemmas

**Lemma E.1** (Lemma 5 in [Alon et al., 2015]). *Let $G = (V, E)$ be a directed graph with $|V| = K$, in which $i \in N^{\mathrm{in}}(G, i)$ for all vertices $i \in [K]$. Assign each $i \in V$ with a positive weight $w_i$ such that $\sum_{i=1}^n w_i \leq 1$ and $w_i \geq \varepsilon$ for all $i \in V$ for some constant $0 < \varepsilon < \frac{1}{2}$. Then*

$$\sum_{i=1}^K \frac{w_i}{\sum_{j \in N^{\mathrm{in}}(G,i)} w_j} \leq 4\alpha(G) \log \frac{4K}{\alpha(G)\varepsilon},$$

*where $\alpha(G)$ is the independence number of $G$.*

**Lemma E.2** (Lemma 9 in [Alon et al., 2013]). *Let $G = (V, E)$ be a directed graph with vertex set $|V| = K$, in which $i \in N^{\mathrm{in}}(G, i)$ for all $i \in [K]$. Let $p$ be an arbitrary distribution over $[K]$. Then, we have*

$$\sum_{i=1}^K \frac{p_i}{\sum_{j \in N^{\mathrm{in}}(G,i)} p_j} \leq mas(G),$$

*where $mas(G)$ is the size of the maximum acyclic subgraphs of $G$.*

