# OpenReview forum: "Practical Contextual Bandits with Feedback Graphs"
_NeurIPS.cc/2023/Conference — NeurIPS 2023 poster_

### Official Review · Reviewer_o1h9 · 2023-06-26

**Soundness:** 3 good
**Presentation:** 3 good
**Contribution:** 3 good
**Rating:** 6
**Confidence:** 4

**Summary:**

This paper studies online learning in a contextual setting when a feedback graph determines the feedback received by the learner. Namely, the learner observes all the losses experienced by the actions in the graph-neighborhood of the action it played.

Feedback graphs are a well-known feedback model for online learning, and their study in the contextual setting is natural. The paper's main result is a general framework (Theorem 3.1) that reduces the problem to the solution at each time step of a convex program. Then the authors instantiate this general theorem for the strongly observable and weakly observable case, obtaining the same regret rates that characterize the non-contextual version of the problem.

The primary technical tool of the paper is the parameter defined in equation (3). Following Foster et al. 2021, the authors use this new formula instead of the bandit one (equation 1) to prove their results.


**Strengths:**

- The problem studied is natural and well-motivated.
- The regret bounds match the lower bounds in Alon et al., 2015, so they are tight.
- The experimental results and the theoretical guarantees support the claim that a richer feedback structure improves the regret bounds w.r.t. the bandit case.
- It is nice that the authors devoted some time to presenting the detailed results for some famous classes of feedback graphs.


**Weaknesses:**

- Once equation 3 is designed, the rest of the paper seems incremental to Foster et al.
- Equation 3 needs knowledge of the feedback graph. Therefore the paper works only in the informed setting.
- The introductory model entails time-varying feedback graphs, while the results for strong and weakly observable feedback graphs need the graph to be deterministic and known up-front.

Minor comment:
- uniform the \citet and \citep notation



**Questions:**

None

**Limitations:**

See above

---

> ### Author Rebuttal · Authors · 2023-08-08
>
> We thank the reviewer for the valuable comments. We address the issues you mentioned as follows.
>
> **1. Once equation 3 is designed, the rest of the paper seems incremental to Foster et al.**
>
> We argue that achieving the tight regret bound with respect to the correct graph-theoretic dependence for different types of feedback graphs requires non-trivial analysis. We refer the reviewer to our response to Reviewer towH for details.
>
> **2. Equation 3 needs knowledge of the feedback graph. Therefore the paper works only in the informed setting.**
>
> Yes, our work focuses on the informed graph feedback setting and we leave the uninformed setting as the future work.
>
> **3. The introductory model entails time-varying feedback graphs, while the results for strong and weakly observable feedback graphs need the graph to be deterministic and known up-front.**
>
> We allow time-varying feedback graphs but the feedback graph needs to be informed at the beginning of each round.  Our algorithm is defined for stochastic graphs, but our analysis focuses on the deterministic graph case in order to create correspondence with classic (non-contextual) minimax bounds for bandits with deterministic feedback graphs.
>
> We will fix the citation notations you mentioned in the next revision.

---

### Official Review · Reviewer_WhSo · 2023-07-05

**Soundness:** 4 excellent
**Presentation:** 3 good
**Contribution:** 3 good
**Rating:** 6
**Confidence:** 4

**Summary:**

The authors consider the adversarial contextual bandit problem with feedback graphs, in the finite function class setting with a realizability assumption, with an access to an online regression oracle. They extend previous approaches for the vanilla contextual MAB setting in order to obtain regret bounds of $\tilde{O}(\sqrt{\alpha T})$ with fully observable graphs and $\tilde{O}(d^{\frac13} T^{\frac23})$ for weakly observable graphs, where $\alpha$ and $d$ denote the graph's independence number and weak domination number respectively. The authors demonstrate their results empirically and show that their algorithm performs better than existing approaches when side observations are available.

**Strengths:**

* The authors exhibit the first efficient algorithm for contextual bandits with feedback graphs, which obtains near-optimal regret guarantees both in the strongly observable and weakly observable settings.
* The techniques utilized by the authors seem to neatly generalize the approach of Foster et al. ('21) to the feedback graph setting, by generalizing the inverse gap technique to solving a convex problem together with utilizing a combinatorial property of the feedback graphs.
* The authors demonstrate in several empirical experiments that their algorithm out-performs SquareCB when feedback graphs are present, thus strengthening the idea that better performance can be obtained when side observations are available.

**Weaknesses:**

* This is not a very major issue, but the algorithm suggested by the authors requires knowledge of the feedback graphs' independence number (or a good bound on it), which is a hard quantity to compute in general. This is only a minor point because many previous works in the feedback graphs literature also require knowing such a parameter, but I will remark that some results can be obtained without this knowledge.


**Questions:**

Given the result presented in the paper, a very natural question that arises is whether or not for stochastic contexts and losses, the approach of [2] can be generalized to the feedback setting in order to obtain similar regret bounds, but with access to an **offline** regression oracle. I think such a result would be also very interesting, and I would expect that similar approaches to those that the authors used in this paper would work for extending [2], as their approach also extends SquareCB of [1]. I'd appreciate it if the authors could comment on whether or not they considered the stochastic setting as well, and if they think that they're approach could be used in order to obtain results with an offline oracle.

[1] Foster, Dylan, and Alexander Rakhlin. "Beyond ucb: Optimal and efficient contextual bandits with regression oracles." _International Conference on Machine Learning_. PMLR, 2020.

[2] Simchi-Levi, David, and Yunzong Xu. "Bypassing the monster: A faster and simpler optimal algorithm for contextual bandits under realizability." _Mathematics of Operations Research_ 47.3 (2022): 1904-1931.

**Limitations:**

Yes

---

> ### Author Rebuttal · Authors · 2023-08-08
>
> We thank the reviewer for the valuable comments. We address the issues you mentioned as follows.
>
> **1. This is not a very major issue, but the algorithm suggested by the authors requires knowledge of the feedback graphs' independence number (or a good bound on it), which is a hard quantity to compute in general. This is only a minor point because many previous works in the feedback graph literature also require knowing such a parameter, but I will remark that some results can be obtained without this knowledge.**
>
> Thanks for pointing this out. In fact, for the strongly observable graph case, we are able to achieve the same regret bound **without** the knowledge of $\alpha_t$ by picking the parameter $\gamma$ in an adaptive way. Specifically, note that for strongly observable graphs, $\gamma\cdot \min_{p\in \Delta_K}\rm\overline{dec_{\gamma}}(p)\leq\widetilde{O}(\alpha_{t})$ at round $t$. Therefore, we start from $\gamma_1=\sqrt{T}$ and keep track of the following value $V_t=\sum_{\tau=1}^t\gamma_{\tau}\cdot \min_{p\in\Delta_K}\rm\overline{dec_{\gamma_\tau}}(p)$, which is indeed obtainable by solving the convex program. When this value is doubled, we double $\gamma_t$ and restart the algorithm; otherwise, we set $\gamma_{t+1}=\gamma_t$. This adaptive tuning does not require the knowledge of $\alpha_t$ and achieve the same $\widetilde{O}\left(\sqrt{\sum_{t=1}^T \alpha_t}\right)$ regret bound with a factor of $\log \left(\sum_{t=1}^T\alpha_t/T\right)$ overhead.
>
> For the weakly observable graph case, since the weak domination number of a graph can be approximated efficiently within a factor of $\log K$, we are able to apply doubling trick to adaptively tune $\gamma$ by approximating the weak domination number directly on the sequence of observed graphs and achieve the same $\widetilde{O}(T^{1/3}(\sum_{t=1}^Td_t)^{1/3})$ regret bound. We will include the adaptive tuning part in the appendix in the next revision.
>
>
> **2. Given the result presented in the paper, a very natural question that arises is whether or not for stochastic contexts and losses, the approach of [2] can be generalized to the feedback setting in order to obtain similar regret bounds, but with access to an offline regression oracle. I think such a result would be also very interesting, and I would expect that similar approaches to those that the authors used in this paper would work for extending [2], as their approach also extends SquareCB of [1]. I'd appreciate it if the authors could comment on whether or not they considered the stochastic setting as well, and if they think that they're approach could be used in order to obtain results with an offline oracle.**
>
> Thanks for pointing this out! In fact, in an ongoing follow-up, we have already had some results with respect to the stochastic context and loss using offline regression oracle. Specifically, we are able to obtain the same $\widetilde{O}(\sqrt{\alpha T})$ regret bound when the graph is self-aware and $\widetilde{O}(d^{1/3}T^{2/3})$ when the graph is weakly observable.

---

> > ### Comment · Reviewer_WhSo · 2023-08-13
> >
> > I thank the authors for their detailed response. After reading the other reviews and comments, I have no further questions about the paper and I'm inclined to leave my score as it is.

---

### Official Review · Reviewer_n8Cc · 2023-07-06

**Soundness:** 3 good
**Presentation:** 3 good
**Contribution:** 2 fair
**Rating:** 6
**Confidence:** 3

**Summary:**

This work is concerned with the problem of contextual bandits with graph feedback. The authors consider a setting where the contexts and the graphs are generated in an arbitrary manner and revealed to the learner at the beginning of each round. For a given context, the mean loss of each arm is assumed to be fixed. Furthermore, it is assumed that the learner is provided with a class of functions mapping context-action pairs to their mean loss, and that this class contains the true function. Following prior works in contextual bandits, the authors use an online regression oracle to estimate the true mean losses. These estimates are then used within the estimation-to-decision framework of  Foster et al. (2021).  Within this framework, the proposed algorithm requires solving a convex program at each round, for which closed form solutions are provided for special cases of interest. The authors prove regret bounds that depend on the structure of the feedback graphs and the regret of the online regression oracle. Additionally, empirical evaluations are carried out to showcase the algorithm's ability to take advantage of the side observations provided via the feedback graphs.

**Strengths:**

Although the adopted approach relies on existing techniques, the application of the estimation-to-decision framework of Foster et al. (2021) for bandits with graph feedback is still an interesting contribution. Most notably, in Theorems 3.2 and 3.4, the authors bound the decision estimation coefficient for their algorithm in terms of the independence number for strongly observable graphs and the weak domination number for weakly observable graphs, which respectively are the graph theoretic quantities that characterize the minimax regret for these settings. Overall, the paper is well written, the setting is adequately motivated, and the clarity of the presentation is decent.

**Weaknesses:**

- The regret bounds in Corollaries 3.3 and 3.5 are stated in terms of a uniform upper bound on the independence number or the weak domination number of the observed graphs. This is unsatisfactory since a single sparse graph could render the bound vacuous.
- In the formulated setting, the graphs are allowed to be stochastic, where each edge is realized with a certain probability which is revealed to the learner at the beginning of the round. However, all the provided theoretical results are for deterministic graphs.
- One point in need of clarification concerning the experiments is that it is mentioned in the beginning of Section 5 that all the graphs used in the experiments are deterministic, while the experiment described in Section 5.2.1 seems to involve stochastic graphs.
- For the experiment of Section 5.1, it might be more informative to include more intermediate cases between bandits and full information.
- A minor correction: In the discussion section, the authors address the limitation of their approach in the uniformed setting, where the feedback graph is only revealed to the learner after making a decision. The authors then cite results for the non-contextual case from (Cohen et al, 2016). However, in that work, the graph is never revealed to the learner, only the actions in the neighbourhood of the played action and their losses are observed.

**Questions:**

- Is it possible to obtain bounds that scale with the average independence number (or weak domination number) of the graphs perhaps via an adaptive choice of the value of gamma at each round?
- What kind of regret bounds can we obtain for stochastic graphs using this approach?
- Does the proposed approach offer an advantage over existing algorithms for learning with graph feedback in the non-contextual case?

**Limitations:**

The authors did address some of the limitations of their work. Notably the fact that their approach requires the knowledge of the feedback graph before choosing the action.

---

> ### Author Rebuttal · Authors · 2023-08-08
>
> We thank the reviewer for the valuable comments. We address the issues you mentioned as follows.
>
> **1. The regret bounds in Corollaries 3.3 and 3.5 are stated in terms of a uniform upper bound on the independence number or the weak domination number of the observed graphs. This is unsatisfactory since a single sparse graph could render the bound vacuous. (Is it possible to obtain bounds that scale with the average independence number (or weak domination number) of the graphs perhaps via an adaptive choice of the value of gamma at each round?**
>
> This is a good point. In fact, we write Corollary 3.3 and 3.5 with respect to the maximum independence number $\alpha$ and maximum weak domination number $d$ only for simplicity, but our regret bound scales with respect to the averaged one $\frac{1}{T}\sum_{t=1}^T\alpha_t$ and $\frac{1}{T}\sum_{t=1}^Td_t$ by applying a doubling trick on the choice of $\gamma$. Specifically, for the strongly observable graph case, we set $\gamma=\sqrt{T}$ initially. If at some round t, $\gamma<\sqrt{\sum_{\tau=1}^t\alpha_{\tau}}$, then we double $\gamma$ and restart the algorithm; otherwise, we keep using the same $\gamma$. This gives the $\widetilde{O}\left(\sqrt{\sum_{t=1}^T\alpha_t}\right)$ regret bound. Applying a similar doubling trick for the weak domination number gives $\widetilde{O}\left(T^{1/3}\left(\sum_{t=1}^Td_t\right)^{1/3}\right)$ regret bound as well. For an improved adaptive tuning method **without even knowing/computing $\alpha_t$ and $d_t$**, we refer to our response (1) for Reviewer WhSo.
>
> **2. In the formulated setting, the graphs are allowed to be stochastic, where each edge is realized with a certain probability which is revealed to the learner at the beginning of the round. However, all the provided theoretical results are for deterministic graphs. (What kind of regret bounds can we obtain for stochastic graphs using this approach?)**
>
> While our algorithm is technically defined for general stochastic graphs, in order to compare our results to the regret bound for standard bandits with feedback graphs, we consider the deterministic case and show that for both strongly observable and weakly observable graph cases, our regret bound matches the minimax rate . For stochastic graphs, if the realized feedback graphs are all strongly observable and $G_t(i,j)$ defines the marginal probability that edge (i,j) is realized, then our algorithm achieves $\widetilde{O}\left(\sqrt{\sum_{t=1}^T\alpha_t}\right)$ regret where $\alpha_t$ is defined as the expected independence number given the stochastic feedback graph $G_t$. Similarly, for weakly observable graphs, our algorithm is able to achieve $\widetilde{O}\left(\left(\sum_{t=1}^Td_t^{1/3}\right)T^{1/3}\right)$ where $d_t$ is the expected weak domination number of the graph. To achieve this bound, we need to tune $\gamma$ in an adaptive way and we refer the reviewer to our response (1) of reviewer WhSo.
>
> **3. One point in need of clarification concerning the experiments is that it is mentioned in the beginning of Section 5 that all the graphs used in the experiments are deterministic, while the experiment described in Section 5.2.1 seems to involve stochastic graphs.**
>
> In Section 5.2.1, we sample a graph from a certain distribution and this (deterministic) graph is informed to the learner at the beginning of each round. Thanks for pointing this out, and we will clarify it.
>
> **4. For the experiment of Section 5.1, it might be more informative to include more intermediate cases between bandits and full information.**
>
> Following the reviewers suggestion, we conducted an additional experiment on dataset RCV1 with the inventory graph (whose independence number is 1) and the averaged regret is shown as follows:
>
> | round:t | averaged regret |
> |---------|-----------------|
> |  10000  |      0.3345     |
> |  20000  |      0.2739     |
> |  30000  |      0.2488     |
> |  40000  |      0.2361     |
> |  50000  |      0.2260     |
>
> which is close to the regret in the full-info and cops-and-robbers cases, showing that the regret indeed scales with the independence number.
>
>
> **5. Does the proposed approach offer an advantage over existing algorithms for learning with graph feedback in the non-contextual case?**
>
> No, it does not. Our approach extends the recent framework of realizable contextual bandit with regression oracle to the more general feedback graph model and achieves the optimal regret bound in both strongly observable and weakly observable graph cases.  Our contribution is focused on the contextual bandit case.  While our approach does achieve minimax rates in the (non-contextual) bandit setting, in that setting, existing minimax-optimal algorithms (eg. Exp3.G in [1]) are available which make less assumptions and are therefore preferable (e.g., non-contextual algorithms do not require realizability as they essentially operate directly in policy space; moveover, non-contextual algorithms exist even for the uninformed graph setting).
>
> Thanks for pointing out the other minor issues in the discussion section and we will fix that in the next revision.
>
> [1] Noga Alon, Nicolo Cesa-Bianchi, Ofer Dekel, and Tomer Koren. Online learning with feedback graphs: Beyond bandits. In Conference on Learning Theory, pages 23–35. PMLR, 2015.

---

> > ### Comment · Reviewer_n8Cc · 2023-08-17
> >
> > Thank you for your response. My evaluation of the paper remains the same; the analysis of the DEC in the feedback graphs setting is an interesting contribution, which would be made more complete with an added discussion on the adaptive choice of gamma and the achievable rates for stochastic graphs.

---

### Official Review · Reviewer_q6E5 · 2023-07-26

**Soundness:** 3 good
**Presentation:** 3 good
**Contribution:** 3 good
**Rating:** 6
**Confidence:** 2

**Summary:**

This work studies the contextual bandits problem in the presence of a feedback graph G_t. An edge (i -> j) in G_t means that taking action a_i allows us to observe the loss for action a_j. The work extends the SquareCB algorithm to this setting, the primary difference being the way the action sampling probability p_t is learned (Eq. (1) vs Eq (4)). The authors also present regret bounds for strongly observable and weakly observable graphs where the bounds improve over the standard contextual bandits setting when the independence number and cardinality of the dominating set is small.

**Strengths:**

 - Overall I found the paper to be well-written where the setup was explained well and the notation was easy to follow.
 - This paper builds on the SquareCB algorithm. I appreciated the fact that the work makes an effort to try to separate their own contributions from the SquareCB paper.
- The paper covers both strongly and weakly observable graphs. In general, the statements in Theorems 3.2 and 3.4 are fairly intuitive.

**Weaknesses:**

 - By and large, the paper relies heavily on the SquareCB paper's analysis. Although the setting of feedback graphs is new, the analysis seems derivative.
- I would suggest that the authors provide more intuition behind the proofs of the key theorems. Even though the theorem statements are easy to understand, it would be nice to get some insight into the proofs, and specifically what are steps different from the standard CB regret analysis.

**Questions:**

 - Are there any intermediate characterizations of graphs between strongly and weakly observable? In the strongly observable case, for nodes without self-loops, you require an edge from all other nodes. What if there are edges from most nodes (but not all)? Or what if most nodes are strongly observable but not all? Broadly, I am trying to understand how conservative the bounds are when the graphs are weakly observable (but perhaps not too weak). Is the algorithm expected to the complexity of the graph somehow?
 - Follow-up to the above, the regret bounds take the worst-case graph over all the rounds. You allow the graph to change over t. Does the regret of your proposed algorithm also improve when the graphs are favorable for a large number of time steps?

(CB is not my area of expertise so I understand that both of the above assumptions are probably standard for doing regret analysis, but I think experimentally it would be nice to get some insight into how the actual algorithm performs relative to the theoretical regret).

**Limitations:**

See weaknesses part.

---

> ### Author Rebuttal · Authors · 2023-08-08
>
> We thank the reviewer for the valuable comments. We address the issues you mentioned as follows.
>
> **1. By and large, the paper relies heavily on the SquareCB paper's analysis. Although the setting of feedback graphs is new, the analysis seems derivative.**
>
> The analysis of SquareCB relies on the constructive inverse gap weighting probability distribution (which is a closed form solution) to bound the DEC term. However, this kind of closed form solution does not exist in the bandit with feedback graph problem. It is even unclear whether this min-max problem can be efficiently solved in the case with feedback graphs. In our analysis, we first show that the min-max problem can indeed be solved efficiently. Then, we use a careful combination of the Sion’s minimax theorem and the graph-theoretic lemma in [1] to bound the DEC term and to achieve the minimax regret for different types of feedback graphs, which is non-trivial.
>
> **2. I would suggest that the authors provide more intuition behind the proofs of the key theorems. Even though the theorem statements are easy to understand, it would be nice to get some insight into the proofs, and specifically what are steps different from the standard CB regret analysis.**
>
> Thanks for the suggestion. We will highlight the differences between SquareCB regret analysis and our regret analysis in the revised version. Furthermore, we will provide additional insights of our proofs, particularly focusing on 1) proving that the min-max problem can be efficiently solved; 2) how we apply the graph-theoretic lemma in bounding the DEC term with feedback graph — a novel aspect that has not been studied before.
>
> **3. Are there any intermediate characterizations of graphs between strongly and weakly observable? In the strongly observable case, for nodes without self-loops, you require an edge from all other nodes. What if there are edges from most nodes (but not all)? Or what if most nodes are strongly observable but not all? Broadly, I am trying to understand how conservative the bounds are when the graphs are weakly observable (but perhaps not too weak). Is the algorithm expected to the complexity of the graph somehow?**
>
> As proven in Theorem 9 of Alon et al. 2015, even if the graph has one weakly observable node, we can construct an instance such that the regret is at least $\mathcal{O}(d^{1/3}T^{2/3})$ (and as shown in Corollary 3.5, our algorithm already achieves this minimax rate).  Regarding relative difficulty: minimax results indicate within each major graph class (strongly/weakly observable graphs) the rate is determined, but the relative difficulty of the problem is affected by an associated graph-theoretic quantity (independence or weak-domination number, respectively) which affects constants.  Our contextual results pleasantly correspond to the existing known non-contextual results.
>
> **4. Follow-up to the above, the regret bounds take the worst-case graph over all the rounds. You allow the graph to change over $t$. Does the regret of your proposed algorithm also improve when the graphs are favorable for a large number of time steps?**
>
> In fact, our regret bound scales with respect to $\sum_{t=1}^T\alpha_t$ ($\sum_{t=1}^Td_t$) in the strongly (weakly) observable case by applying a doubling trick on the choice of $\gamma$. We refer the reviewer to our response for question 1 of Reviewer n8Cc and Reviewer WhSo for details on how to tune $\gamma$. Therefore, when the graphs are favorable, i.e., the independence number is small, our regret bound also improves upon the one with the worst-case graph.
>
>
> [1] Noga Alon, Nicolo Cesa-Bianchi, Ofer Dekel, and Tomer Koren. Online learning with feedback graphs: Beyond bandits. In Conference on Learning Theory, pages 23–35. PMLR, 2015.

---

> > ### Comment · Reviewer_q6E5 · 2023-08-13
> > **Response**
> >
> > I thank the authors for their response. My (positive) evaluation of the paper remains the same.

---

### Official Review · Reviewer_towH · 2023-07-28

**Soundness:** 3 good
**Presentation:** 3 good
**Contribution:** 3 good
**Rating:** 5
**Confidence:** 3

**Summary:**

This work studied contextual bandits with feedback graphs. The authors provided an algorithm based on the recent Decision-Estimation Coefficient (DEC) framework that finds the next-step action distribution by solving a minimax optimization problem. To address the issue that the minimax problem is hard to solve, the authors also showed that there exists an efficient implementation of the solver, and the closed-form solution also exists for some special cases. For the experiment, the authors compared the proposed algorithm with a vanilla baseline algorithm that does not utilize the feedback graph information.

**Strengths:**

The presentation is clear. The overall approach to solving the contextual bandits with feedback graph is convincing. The theoretical results are sound.



**Weaknesses:**

The importance of this work remains unclear. The proposed algorithm is very similar to the original E2D algorithm proposed by Foster et al. 2021 with an additional expectation over the feedback graph. The analysis technique is also very similar. The key difficulty in proving the regret of SquareCBG is to build an effective estimate the DEC constant. However, from the proof of Theorem 3.2, it seems that the proof is pretty standard by following the decomposition technique developed in Foster et al. 2021 (like their Proposition 5.1), while utilizing the graph node estimation lemma proposed by Alon et al. 2015. Therefore, I would recommend the authors highlight the main challenge for them to derive the theoretical results.


A typo. Line 62, for some action $j$-> $a_t$.




**Questions:**

See Weaknesses.

**Limitations:**

The authors addressed the limitations.

---

> ### Author Rebuttal · Authors · 2023-08-08
>
> We thank the reviewer for the valuable comments. We address the issues you mentioned as follows.
>
> **1. The importance of this work remains unclear. The proposed algorithm is very similar to the original E2D algorithm proposed by Foster et al. 2021 with an additional expectation over the feedback graph. The analysis technique is also very similar. The key difficulty in proving the regret of SquareCB.G is to build an effective estimate of the DEC constant. However, from the proof of Theorem 3.2, it seems that the proof is pretty standard by following the decomposition technique developed in Foster et al. 2021 (like their Proposition 5.1), while utilizing the graph node estimation lemma proposed by Alon et al. 2015. Therefore, I would recommend the authors highlight the main challenge for them to derive the theoretical results.**
>
> We agree that our graph-based DEC term is inspired by the work of [1]. However, the E2D algorithm is defined for general sequential learning problems and it is unclear whether this is an **efficient** algorithm for learning  for contextual bandit with **general graph** feedback due to the complexity of solving the min-max problem. In [1], all the instances of the DEC upper bound is derived in the bandit feedback case while we propose the **first efficient algorithm** for contextual bandit with general graph feedback with optimal regret bound. This is achieved by first showing that the minimax problem can be solved in an efficient way and then proving that DEC is well-bounded with respect to corresponding graph-theoretic numbers. Although the graph-theoretic lemma in [2] is an existing technique, how to apply it to bounding the specific DEC term for contextual bandit with feedback graph is unclear and requires non-trivial analysis. The analysis for contextual bandit in [1] relies on the constructive inverse gap weighting probability distribution, which is a closed form solution to the min max problem. However, this kind of closed form solution does not exist in our problem. Instead, in our analysis, we use a careful combination of Sion’s minimax theorem and the graph-theoretic lemma and achieve minimax regret bound $\widetilde{\mathcal{O}}(\sqrt{\alpha T})$ for strongly observable graphs. Moreover, we analyze the weakly observable graphs, where the available feedback is less informative than the bandit feedback. In this scenario, we also achieve the minimax regret bound $\widetilde{\mathcal{O}}(d^{1/3}T^{2/3})$.
>
> Notably, we also want to highlight that our contribution is not limited to the theoretical side but our algorithm is practical and can be implemented efficiently in practice. Our experimental results prove the effectiveness of our approach.
>
> **2. A typo. Line 62, for some action $j\rightarrow a_t$**
>
> This is not a typo. In Line 62, we consider the stochastic feedback graph case. Given the selected action $a_t$ and the stochastic feedback graph $G_t$, we observe the loss of action $j$ with probability $G_t(a_t,j)$.
>
> [1] Dylan J Foster, Sham M Kakade, Jian Qian, and Alexander Rakhlin. The statistical complexity of interactive decision making. arXiv preprint arXiv:2112.13487, 2021.
>
> [2] Noga Alon, Nicolo Cesa-Bianchi, Ofer Dekel, and Tomer Koren. Online learning with feedback graphs: Beyond bandits. In Conference on Learning Theory, pages 23–35. PMLR, 2015.

---

### Author Rebuttal · Authors · 2023-08-08

We thanks all the reviewers for their valuable comments, especially for pointing out the adaptive tuning issue of the parameter of $\gamma$ without requiring knowledge of the graph-theoretic quantities and beyond the worst-case graph. We address your issues in separate sections as follows.

---

### Decision · Program_Chairs · 2023-09-21

**Decision:**

Accept (poster)

**Comment:**

This paper studies the CB problem with stochastic feedback graphs in the informed setting. The authors propose a novel algorithm based on the SquareCB ideas. Reviewers find that the proposed approach is novel and that the paper has a good presentation. Multiple reviewers have expressed concern about the technical novelty of the proposed approach and in particular have noted that a large part of the proof follows the established approach in Foster et al. 2021. I find that the authors' rebuttal addresses the stated concerns. Multiple reviewers also note the issue with tuning $\gamma$. I find the rebuttal to address this issue, however, the authors should include the detailed derivation for the doubling trick in the final version of the paper.